# Dynamic assembly of a large multidomain ribozyme visualized by cryo-electron microscopy

Shekhar Jadhav[1,2], Mauro Maiorca[3,4,5,10], Jacopo Manigrasso[6,9,10], Spandan Saha[2], Auriane Rakitch[2], Stefano Muscat[6], Thomas Mulvaney[4,5], Marco De Vivo[6] ✉, Maya Topf[3,4,5] ✉ & Marco Marcia[1,2,7,8] ✉

Many RNAs rely on their 3D structures for function. While acquiring functional 3D structures, certain RNAs form misfolded, non-functional states ('kinetic traps'). Instead, other RNAs sequentially assemble into their functional conformations over pre-folded scaffolds. Elucidating the principles of RNA sequential assembly is thus important to understand how RNAs avoid the formation of misfolded, non-functional states. Integrating single-particle electron cryomicroscopy (cryo-EM), image processing, in solution small-angle X-ray scattering (SAXS), EM-driven molecular dynamics (MD) simulations, structure-based mutagenesis, and enzymatic assays, we have visualized the sequential multidomain assembly of a self-splicing ribozyme of biomedical and bioengineering significance. Our work reveals a distinct dynamic interplay of helical subdomains in the ribozyme's 5'-terminal scaffold, which acts as a gate to control the docking of 3'-terminal domains. We identify specific conserved and functionally important secondary structure motifs as the key players for orchestrating the energetically inexpensive conformational changes that lead to the productive formation of the catalytic pocket. Our work provides a near-atomic resolution molecular movie of a large multidomain RNA assembling into its functionally active conformation and establishes a basis for understanding how RNA avoids the formation of non-functional 'kinetic traps'.

RNA catalyzes vital biochemical reactions, such as splicing or protein synthesis, and regulates gene expression and metabolic adaptation to the environment, with important implications for human health[1–4]. To enable such fundamental processes, RNA molecules often need to assemble into functional three-dimensional (3D) structures by exploring complex conformational landscapes[5,6]. RNA structural assembly can occur via two main pathways[7], known as the 'kinetic trap' and the 'sequential' folding pathways.

Frequently, RNA folding involves the formation of stable misfolded states, i.e., so-called 'kinetic traps', which are functionally

[1]European Molecular Biology Laboratory (EMBL) Grenoble, 71 Avenue des Martyrs, Grenoble, France. [2]Department of Cell and Molecular Biology, Uppsala University, Husargatan 3, Uppsala, Sweden. [3]Centre for Structural Systems Biology (CSSB), Hamburg, Germany. [4]Research Department Integrative Virology, Leibniz-Institut für Virologie (LIV), Hamburg, Germany. [5]Institute for Molecular Virology and Tumorvirology, Universitätsklinikum Hamburg Eppendorf (UKE), Hamburg, Germany. [6]Laboratory of Molecular Modelling & Drug Discovery, Istituto Italiano di Tecnologia, Via Morego 30, Genoa, Italy. [7]Istituto Italiano di Tecnologia, Via Morego 30, Genoa, Italy. [8]Science for Life Laboratory, Department of Cell and Molecular Biology, Uppsala University, Husargatan 3, Uppsala, Sweden. [9]Present address: Medicinal Chemistry, Research and Early Development, Cardiovascular, Renal and Metabolism (CVRM), BioPharmaceuticals R&D, AstraZeneca, Gothenburg, Sweden. [10]These authors contributed equally: Mauro Maiorca, Jacopo Manigrasso. ✉e-mail: marco.devivo@iit.it; maya.topf@cssb-hamburg.de; marco.marcia@icm.uu.se

unproductive[8–11]. For instance, tRNAs form misfolded states that cannot be recognized by their cognate aminoacyl transferases unless denatured and refolded[12,13]. Ribosomal RNAs form alternative stable conformations with different abilities to bind ribosomal proteins and thus reconstitute a functional ribosome[14]. The M1 catalytic subunit of *Escherichia coli* ribonuclease P forms a transiently inactive conformation that needs to refold before it can catalyze its reaction[15]. Finally, certain hammerhead ribozymes form kinetically-trapped conformations that have low catalytic efficiency[16]. Mechanistically, the process of kinetic trap formation is best studied in group I introns, which are self-splicing ribozymes that fold, generating a variety of misfolded species[7]. The structural basis for the formation of 'kinetic traps' in these model systems has been recently elucidated thanks to the determination of a number of cryo-EM structures of the *Tetrahymena* group I intron in the catalytically active and in misfolded states[9,17–20].

In some cases, though, RNAs are able to fold sequentially into their functional structures, without forming 'kinetic traps'. These latter RNAs crucially form stable, well-folded core scaffolds that guide the sequential docking of peripheral RNA domains. Elucidating how RNAs can mechanistically avoid the formation of 'kinetic traps' would open new avenues for deriving quantitative models for predicting RNA structures and RNA thermodynamic behaviors, with important impacts on RNA engineering and related technological developments. However, to date, we still lack a direct visualization and a mechanistic model of how RNAs can assemble sequentially, avoiding misfolded, non-functional states[8,9].

The prototypical systems used to study sequential RNA assembly are group II introns, another class of highly-conserved self-splicing ribozymes of biomedical and bioengineering relevance[21]. Group II introns are validated drug targets for antifungal antibiotics, potentially powerful vectors for genetic engineering, and the evolutionary ancestors of the catalytic RNA core of the eukaryotic spliceosome[22,23]. Notably, from biochemical work on the yeast *ai5γ* group II intron from *Saccharomyces (S.) cerevisiae*, it is well established that the 5′-terminal domain 1 (D1, ~300 nucleotide length, ~90 kDa) first folds autonomously into a stable scaffold. Then, folding of D1 is followed by the assembly of 3′-terminal domains (D2-D5), without generating kinetic traps, to eventually form highly-conserved tertiary interactions that stabilize the catalytically-active structure, formed by domains D1-5[24–27]. Although mechanistically unclear, this assembly process is highly conserved in evolution, throughout all classes of group II introns[28].

The crystal structure of D1 from the bacterial I1 group II intron of *Oceanobacillus (O.) iheyensis* confirmed the formation of the D1 assembly scaffold as the first step for intron folding[28]. Nevertheless, comparison of the D1 structure with structures of D1-5 at various catalytic stages[29,30] revealed that in isolation D1 adopts a "closed" conformation unable to accommodate in its core the active site motifs, provided by D2-D5[28]. This observation raises the question of how D1 opens its core without acting as a 'kinetic trap', and how it adopts its "catalytic" conformation in the full-length functionally active ribozyme, guided by the assembly of D2-D5. The mechanism regulating the conformational changes of D1, the energetics of this transition, and the hierarchy by which D2-D5 dock on D1 still remain unknown.

Here, to shed light on the molecular mechanism of group II intron assembly, we have set out to visualize the D1 dynamics in the presence of D2, D2 and D3, and D2, D3, and D4, respectively. We have integrated electron cryo-microscopy (cryo-EM), single-particle image analysis, in-solution small-angle X-ray scattering (SAXS), and molecular dynamics (MD) simulations. Our data reveal conformational dynamics of conserved D1 hinge residues, which were not apparent from the existing crystal structures of the group II intron ribozyme. These hinge residues emerge as the structural determinants that enable group II intron assembly by acting as a gate to regulate the energetically inexpensive formation of the active site. Our proposed 'dynamic gating' mechanism is evolutionarily conserved across group II introns, because

structure-based mutagenesis aimed to rigidify the D1 hinge residues causes severe catalytic defects on homologous bacterial and yeast introns. Our mechanism thus sets a paradigm to understand how large multi-domain RNAs assemble, avoiding non-functional 'kinetic traps'.

## Results

### Specific helical motifs in group II intron domain 1 (D1) control accessibility to the active site pocket

To understand the mechanism by which the group II intron domain D1 transitions between its "closed" and its "catalytic" conformations, we first set out to identify the critical structural differences between these two states by comparing the corresponding crystal structures. We have compared PDB id 4Y1O (isolated D1 in the "closed" state) and PDB id 4FAQ (D1-5 intron immediately preceding the first step of splicing, where D1 is in the "catalytic" state)[28,29]. Comparison of pseudo-torsional angles of D1 residues in these two structures had previously identified two hinge motifs, formed by residues 71–73 and 115–116 (hinge 1) and residues 217 and 223–224 (hinge 2), respectively, as the control hubs regulating D1 dynamics[28] (Fig. 1A).

Through closer inspection of pairwise phosphate atom distances, here we have quantified the magnitude of structural displacement of each residue between the "closed" and "catalytic" states. We superposed "closed" D1 over "catalytic" D1 by performing an all-atom alignment and then calculating pairwise phosphate distances between analogous phosphate atoms in the two structures, in PyMOL (v 1.8.0.4, Schrodinger LLC). The least structurally-displaced residues correspond to nucleotides 124–133 and 230–242 forming the so-called D1d1 subdomain (see the group II intron secondary structure map in Fig. 1A). By contrast, the most structurally-displaced residues correspond to nucleotides 6–23 and 250–267 in $i_1$-$i_2$ and to nucleotides 76–111 in D1c, around hinge 1 (Fig. 1A and Supplementary Fig. 1A, B). At this location, we have further observed that residues A72 and C116 in hinge 1 change conformation around the phosphate-ribose backbone between the two states. In the "closed" state, the nucleobases of A72 and C116 bulge outwards with respect to the helical axis of D1c. By contrast, in the "catalytic" state, these nucleobases are oriented towards the center of the helix (Fig. 1I). Importantly, in the bulged-out conformation, A72 displays a steric clash with the so-called Z-anchor and λ motifs, inducing them to be unstructured in the "closed" state (Fig. 1J). These unstructured residues define a boundary between two subdomains of D1c. The D1c subdomain closer to the D1 core (hereafter 'core D1c') encompasses residues 66–75 and 112–121, while the more peripheral subdomain (hereafter 'peripheral D1c') encompasses residues 80–102 (Fig. 1 and Supplementary Fig. 1). The relative position of 'core D1c' and 'peripheral D1c', and their orientation with respect to neighboring subdomains $i_1$-$i_2$ and D1d1 crucially determine the overall geometry of the D1 scaffold and the accessibility of its inner cavity to active site nucleotides.

To quantitatively describe the geometry of the D1 scaffold and in line with previous literature discussing conformational changes in biological macromolecules[31–33], we have defined the following specific descriptors. '*Angles A-D*' are the angles between the helical axes of 'core D1c' and D1d1, 'core D1c' and 'peripheral D1c', 'core D1c' and $i_1$-$i_2$, and D1d1 and $i_1$-$i_2$, respectively. '*Gate distance*' is the interatomic distance between the C1′ atoms of two residues (G75 and U238) in D1 and serves as a measurement for the size of the D1 pocket (Fig. 1B–F and Supplementary Table 1). We have then monitored how these angles and distances vary between the "closed" and "catalytic" states of D1. Our analysis shows that in the "closed" state 'core D1c' is closer to D1d1 than in the "catalytic" state, while 'peripheral D1c' displaces the laterally stacked $i_1$-$i_2$ helix. As a consequence, in the "closed" state, the $i_1$-$i_2$ helix occupies a position that would clash with D5 residues in the "catalytic" state, creating a steric hindrance that prevents the "closed" D1 from accommodating D5 into the active site pocket (Fig. 1G, H, and Supplementary Movie 1).

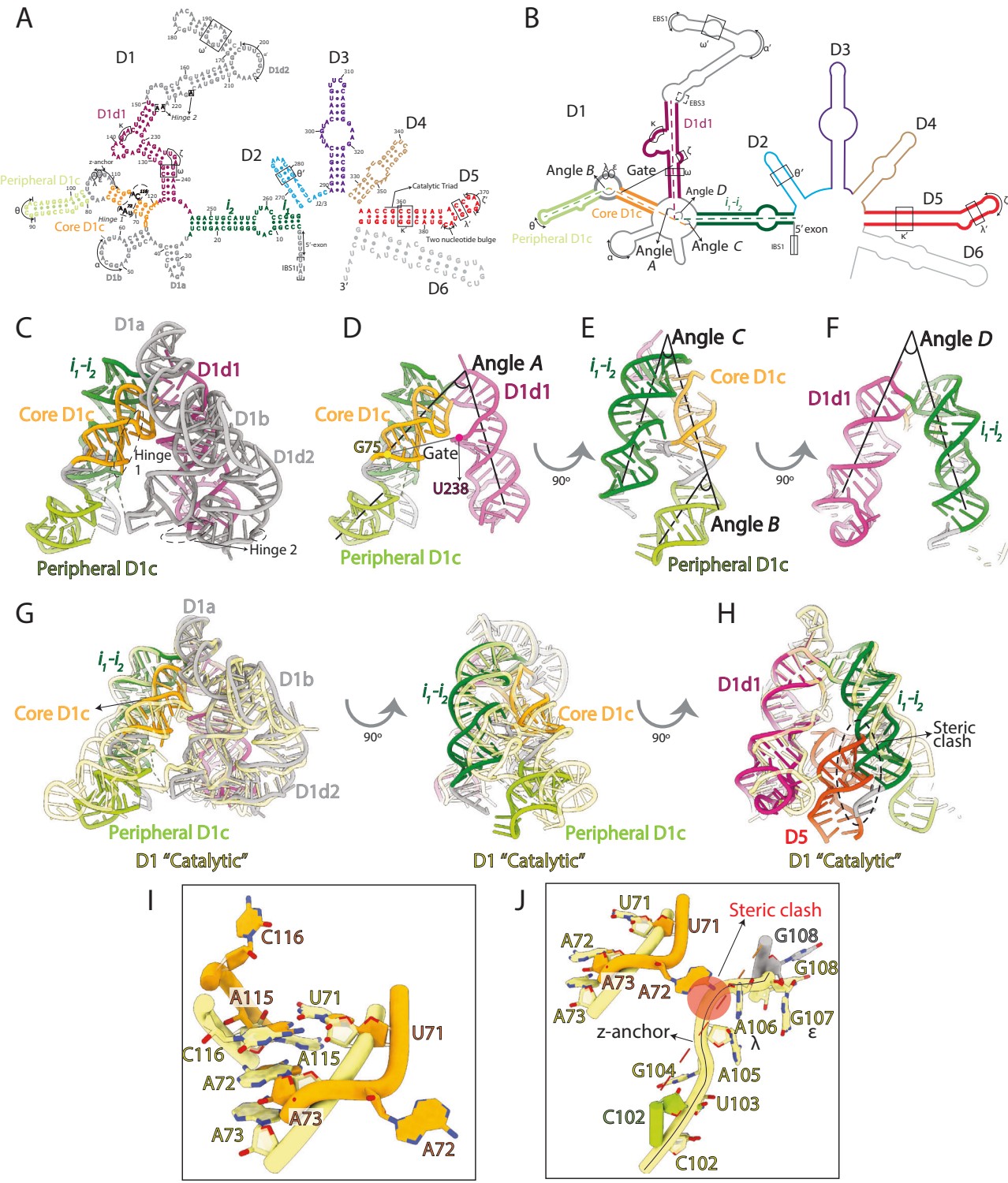

## In solution, domain 1 (D1) is stabilized in a "closed" conformation

To experimentally characterize the absolutely required transition of D1 between its "closed" and its "catalytic" state, we have first studied the behavior of D1 in solution. We have produced and purified D1 by in vitro transcription followed by our previously described non-denaturing RNA purification protocol [see Materials and Methods and refs. 28,34]. We have obtained a pure and homogeneous preparation of D1 as judged by agarose gel electrophoresis, size-exclusion chromatography (SEC), and mass photometry (MP, Supplementary Fig. 2). Size-exclusion chromatography coupled to SAXS (SEC-SAXS) further confirmed that purified D1 in

solution has dimensions and a conformation similar to the ones captured by its crystal structure (PDB id 4Y1O, $X^2_{crysol} = 1.77$, Supplementary Fig. 3 and Supplementary Tables 2, 3). To characterize the dynamics of D1 and the bulged-in *vs* bulged-out conformations of nucleotides A72 and C116 between the "catalytic" and "closed" state, we have then performed SAXS-driven metainference metadynamics simulations. The MD-generated conformational ensemble of D1 agrees with the experimental hydrodynamic behavior of D1 determined by SEC-SAXS ($X^2_{MD} = 1.62 \pm 0.42$, Supplementary Fig. 4). Our simulations also estimated that the Gibbs free energy required to exchange between the bulged-in and bulged-out conformations of A72 and C116 is -10 kcal•mol⁻¹, which is in line with

**Fig. 1 | Group II intron structural motifs. A** Secondary structure map of the *O. iheyensis* I1 group II intron. Domains are abbreviated as D1 to D5. Secondary structure elements of D1 described in the text are depicted with respective labels and colors corresponding to all subsequent subpanels. Tertiary interactions and active site motifs are depicted in Roman letters. **B** Same map highlighting key RNA helices and geometric descriptors used in the text, i.e., 'angles A-D' and 'gate distance'. **C** D1 crystal structure (PDB id 4Y1O) in which each subdomain discussed in the text is represented with a distinctive color, as in panel A. The position of the hinge 1 and 2 residues is indicated. **D** Same structure as in (**C**), where the helical axes of 'core D1c' (orange) and D1d1 (violet), used to calculate 'angle A', are indicated by the black lines. **E** Same structure as in (**C**), where the helical axes of 'core D1c' (orange), 'peripheral D1c' (light green) and $i_1$-$i_2$ (dark green), used to calculate 'angle

B' and 'angle C', respectively, are indicated by the black lines. **F** Same structure as in (**C**), where the helical axes of D1d1 (violet) and $i_1$-$i_2$ (dark green), used to calculate 'angle D', are indicated by the black lines. **G** Superposition of "closed" D1 (PDB id 4Y1O, colored as in panel **C**) and "catalytic" D1 (PDB id 4FAQ, light yellow). **H** Same representation as in (**G**), rotated by 90° with respect to the right subpanel **G**, and showing here the steric clash between the "closed" D1 structure and D5. **I)** Zoom into the hinge 1 residues from "closed" D1 (PDB id 4Y1O, orange) and "catalytic" D1 (PDB id 4FAQ, light yellow). **J** Highlight into the steric clash (red label) between A72 (orange, "closed" state) and nucleotides of the Z-anchor motif (yellow, "catalytic open" state). Z-anchor nucleotides are flexible in the "closed" state. In this latter state, the first nucleotides flanking the Z-anchor and visible in the structure are C102 (light green) and G108 (grey).

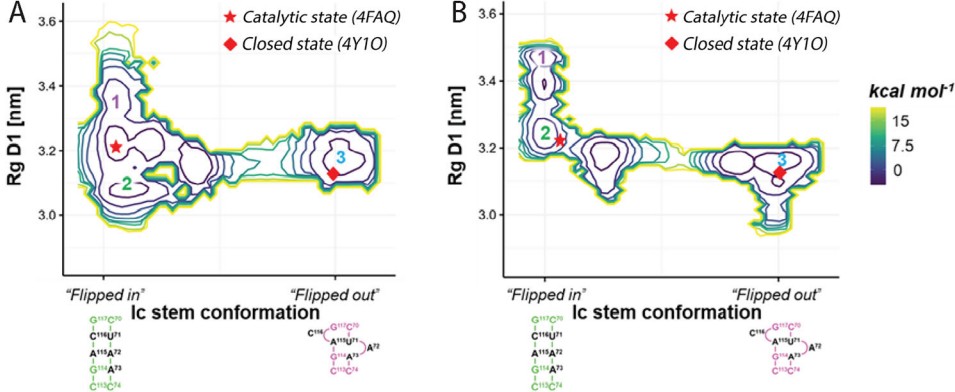

**Fig. 2 | Structural dynamics of D1 as a function of the hinge 1 conformations. A** Free energy surface estimated with SAXS-driven metainference metadynamics simulations for isolated D1 (PDB id 4Y1O). **B** Free energy surface estimated with SAXS-driven metainference metadynamics simulations for D1-2 (coordinates extracted from PDB id 4FAQ). In both panels, the free energy surface is reported as

a contour plot every 2 kcal·mol$^{-1}$. The x-axes report the distance of the simulated system from the A72 and C116 bulged in/out conformations, which are shown as 2D structures below the axes. The y-axes report the $R_g$ of D1. Reference "catalytic" and "closed" states are represented as a star and a square, respectively.

---

similar conformational changes observed previously for catalytic intermediates of the same ribozyme[30] and much smaller than conformational transitions required to refold kinetically trapped RNA conformers[14–16]. Most importantly, simulations show that the conformation of A72 and C116 strongly affects the structural dynamics of D1. When A72 and C116 are bulged-in, D1 is flexible and explores two metastable conformations in one main free-energy basin corresponding to the "catalytic" state (Fig. 2A, states 1 and 2, $X^2 = 1.09$ and $X^2 = 1.83$, respectively). Instead, when both A72 and C116 are bulged out of the D1c helix, D1 only explores "closed" conformations (Fig. 2A, state 3, $X^2 = 1.93$).

Then, we analyzed how the metastable states observed during D1 dynamics differ in terms of the critical structural angles and distances that define the position of the D1 helices ('angles A-D' and 'gate distance', see previous paragraph, Supplementary Fig. 4). From state 1 to state 3, both 'angle A' and 'angle D' gradually decrease. This observation suggests that the dynamics of both 'core D1c' and $i_1$-$i_2$ contribute to the opening and closing of the D1 core, as shown also by the progressive narrowing of the 'gate distance' from state 1 to state 3. Furthermore, 'angle B' gradually increases, revealing a change in the angular orientation of the 'peripheral D1c' helix with respect to the 'core D1c' helix (Supplementary Fig. 4).

In summary, in isolation, the D1 scaffold is stabilized in the "closed" conformation, as captured by the corresponding crystal structure, because A72 and C116 in hinge 1 adopt a bulged-out state, which positions 'peripheral D1c' and $i_1$-$i_2$ in an orientation that closes the D1 core.

### Addition of domain 2 (D2) onto D1 triggers the opening of the D1 cavity

Interestingly, in the "closed" state, 'peripheral D1c' is located in the space that is occupied by D2 in the "catalytic" state (Supplementary Fig. 1). Thus, we went ahead to determine whether the addition of D2 to D1 could facilitate the transition from the "closed" to the "catalytic" state.

Using the same non-denaturing purification protocol as for D1[28,34], we obtained pure, folded, and homogeneous preparations of D1-2, as confirmed by gel electrophoresis, SEC, MP, and SEC-SAXS (Supplementary Fig. 2, 3 and Supplementary Tables 2, 3). Then, we used the experimental scattering data to perform SAXS-driven metainference metadynamics simulations (using a model structure of D1-D2, extracted from the structure of the "catalytic" state, i.e., PDB id: 4FAQ). These simulations agreed with the properties of D1-2 experimentally observed in solution by SEC-SAXS ($X^2_{MD} = 1.70 \pm 0.46$, Supplementary Fig. 4 and Supplementary Tables 2, 3), and estimated a free energy barrier of ~12 kcal·mol$^{-1}$ between the bulged-in and bulged-out conformations of hinge 1, similar to the barrier we observed when simulating D1 in isolation (see above). Importantly, simulations showed that the inclusion of D2 still allows D1 to explore three metastable states, but with an altered conformational landscape suggesting that D2 modifies the structural dynamics of D1. Indeed, in the presence of D2, when A72 and C116 are bulged-out, D1 only samples the "closed" state, as for D1 in isolation (state 3, $X^2 = 2.20$, Fig. 2B). But when A72 and C116 are bulged-in, D1 samples both the "catalytic" state (state 2, $X^2 = 1.23$) and a distinct state with larger $R_g$ that is not sampled by D1 in isolation (state 1, $X^2 = 1.67$, Fig. 2B). These dynamics are quantitatively described by the D1 descriptors, 'angles A-D' and 'gate distance' (Supplementary Fig. 4).

In summary, our SAXS-driven MD simulations suggest that D2 is both required and sufficient to induce D1 to sample "open" states, not explored by D1 in isolation.

### D1 acquires previously-unpredictable conformations in the presence of D2, D3, and D4

To validate the mechanistic hypothesis suggested by SAXS-driven MD simulations and visualize the putative D1 state characterized by larger

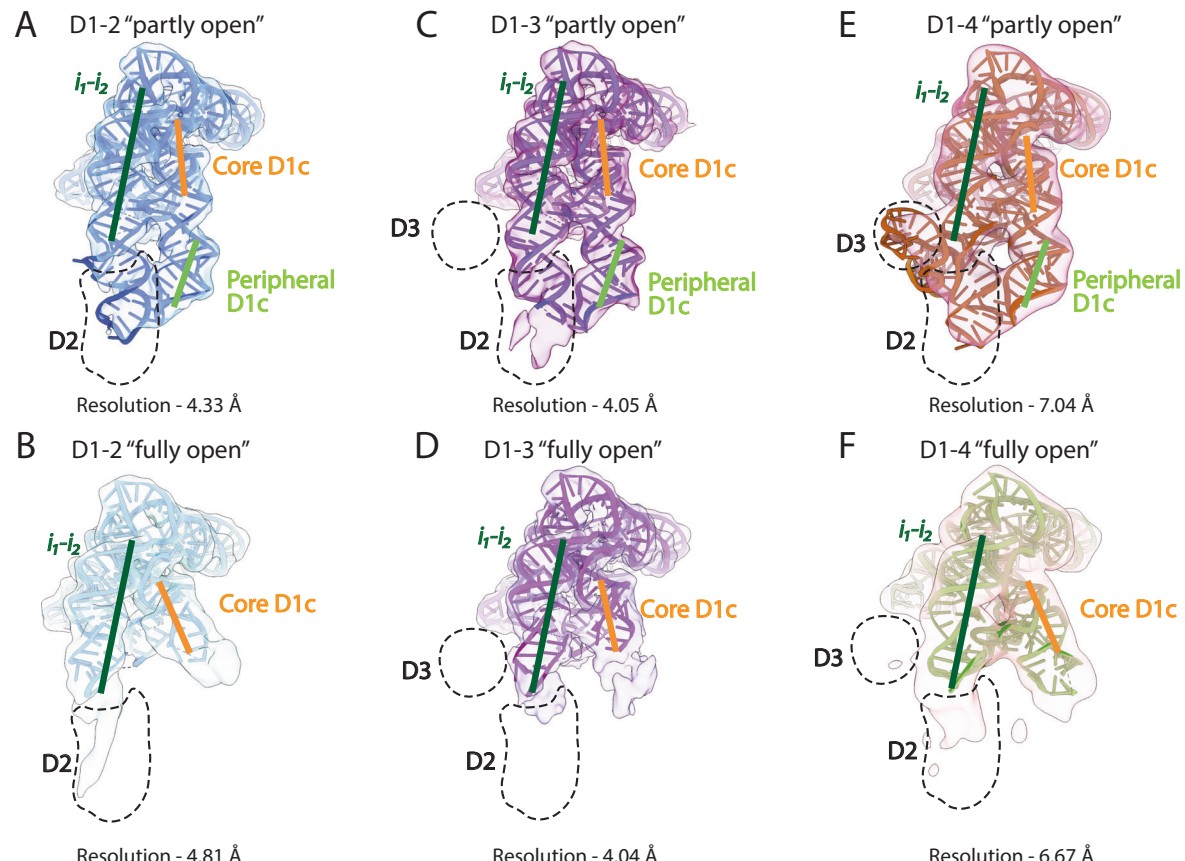

**Fig. 3 | Scaffold of D1 in the presence of D2, D2-3 and D2-3-4. A** Coulomb potential density map of the "partly open" state of D1-2 at 4.33 Å resolution. **B** Density of the "fully open" state of D1-2 at 4.81 Å resolution. **C** Density of the "partly open" state of D1-3 at 4.05 Å resolution. **D** Density of the "fully open" state of D1-3 at 4.04 Å resolution. **E** Density of the "partly open" state of D1-4 at 7.05 Å resolution. **F** Density of the "fully open" state of D1-4 at 6.67 Å resolution. In all panels, thick lines represent the helical axes of 'core D1c' (orange), 'peripheral D1c' (light green), and $i_1$-$i_2$ (dark green), respectively. The expected position of D2 and D3 is indicated with a black dashed encircled region.

$R_g$ values, we set out to experimentally determine the 3D structure of D1-2 using cryo-EM. To obtain a complete picture of the multi-domain assembly mechanism of group II introns, we also determined the cryo-EM structures of D1-3 and D1-4, which we could purify using analogous non-denaturing purification protocols[28,34], to the same degree of purity and homogeneity as D1 and D1-2 (Supplementary Fig. 2). All these assembly intermediates are catalytically inactive as they lack the active site residues contributed by D5[21].

Through SAXS studies, we first confirmed that all constructs are homogenous in solution and suitable for cryo-EM grid vitrification (Supplementary Fig. 3 and Supplementary Tables 2, 3). Vitrified grids displayed sufficiently homogenous particle distribution and particles density on the micrographs to enable particle picking, although a tendency to form more aggregates and sparser particles was observed for the larger constructs (Supplementary Figs. 5–7).

Through 3D reconstruction of 300,000-600,000 particles for each data set in *CryoSPARC*[35–37], we obtained Coulomb potential maps at ~4 Å resolution for D1-2 and D1-3, enabling us to distinguish the major and minor groves of the RNA helices, and at ~7 Å resolution for D1-4, likely correlating to the inherent flexibility of this construct (Fig. 3). These datasets enabled us to properly orient the D1 coordinates into all resulting maps and to identify the D1 subdomains, using the available D1 crystal structures as reference models (Supplementary Figs. 5–7).

Importantly, subsequent 3D classification yielded two distinct cryo-EM maps for each construct, which differ in the relative orientation of the 'core D1c' and the $i_1$-$i_2$ helices (Fig. 3). Surprisingly, in these

constructs, D1 never forms the "closed" or the "catalytic" conformations, but rather adopts two distinct conformations (see values of 'angles A-D' and 'gate distance' in Supplementary Table 1). One conformation, hereafter referred to as the "partly open" state, displays a distinct orientation of 'peripheral D1c', which differs clearly from the "catalytic" state despite laterally flanking $i_1$-$i_2$ and D2 (Supplementary Fig. 8). The other conformation, hereafter referred to as the "fully open" state, displays an even larger separation between D1c and $i_1$-$i_2$, a weak density for the 'peripheral D1c' motif, and disordered Z-anchor and θ nucleotides, which engage in tertiary contacts between D1 and D2 (Supplementary Fig. 8).

In summary, our cryo-EM data revealed that in the presence of 3'-terminal domains, D1 adopts previously uncharacterized conformations characterized by distinct orientations of RNA helices. These conformations originate from the relative displacement of the two D1 helical subdomains, D1c and $i_1$-$i_2$.

## The accessibility of the D1 cavity is regulated by the continuous dynamic angular motion of the D1c helix

To better characterize the conformational space sampled by D1 in the presence of 3'-terminal domains, we have employed 3D variability (3DVA) classification on the particles extracted from the D1-2, D1-3, and D1-4 datasets (Supplementary Fig. 9). We have generated 20 sub-states for each construct. Based on the angular orientation of the D1c and $i_1$-$i_2$ helices, we defined sub-state 1 as the state corresponding to the "partly open" state, and sub-state 20 as the state corresponding to the "fully open" state.

We then refined the structural coordinates of D1 into each of the 3DVA sub-states for the D1-2 and D1-3 datasets. We obtained models characterized by accurate geometry and fit to the respective cryo-EM maps, although some peripheral nucleotides in each construct are structured only in certain, but not all sub-states (Table 1 and Supplementary Figs. 10–12). For the D1-4 dataset, we could not obtain a resolution better than 5 Å from 3DVA. Thus, despite unequivocally observing a progressive separation between D1c and $i_1$-$i_2$ helices from sub-states 1 to 20, we could not visualize further structural details with sufficient confidence for this construct (Supplementary Fig. 7).

Notably, the resulting 3DVA conformational ensembles of D1-2 and D1-3 fit the respective SAXS scattering curves with an improved $X^2$ value compared to the fits obtained using the crystal structures of D1 or the MD computational ensembles ($X^2_{3DVA} = 1.26$ for D1-2, and $X^2_{3DVA} = 1.04$ for D1-3, Supplementary Tables 2, 3). We then selected three parameters to describe the conformation of D1 in each sub-state, namely the pairwise root mean square deviation (RMSD) value of the models, and the previously-defined 'angle A' and 'gate distance'. For reference, the RMSD between the "closed" and "catalytic" conformations of D1 is 2.4 Å. Moreover, in the "closed" conformation of D1 (PDB id 4Y1O), 'angle A' = 62.9° and 'gate distance' = 32.0 Å, while in the "catalytic" conformation of D1 (PDB id 4FAQ), 'angle A' = 57° and 'gate distance' = 34.0 Å.

The pairwise RMSD value ranged between 0.1 to 1.2 Å for the 20 sub-states of D1-2, and between 0.1 to 0.5 Å for the 20 sub-states of D1-3 (Fig. 4A, B). These values suggest that D1-3 undergoes less pronounced conformational dynamics compared to D1-2. Furthermore, 'angle A' progressively increases from 65.8° for sub-state model 1 to 80.5° for sub-state model 20 for the D1-2 dataset, and from 66.9° for sub-state model 1 to 75.1° for sub-state model 20 for the D1-3 dataset, respectively (Fig. 4C). Finally, the 'gate distance' progressively increases from 33.8 Å for sub-state model 1 to 37.8 Å for sub-state model 20 for the D1-2 dataset, and from 34.6 Å for sub-state model 1 to 36.5 Å for sub-state model 20 for the D1-3 dataset, respectively (Fig. 4D). These displacements correspond to movements at the tip of helices D1c and D1d1 of 51.2 Å (Figs. 5, 6).

In summary, 3DVA analysis confirms the observations made during heterogeneous-refinement 3D classification. In the presence of 3'-terminal domains, the D1 conformation is significantly different from the "closed" state adopted by D1 in isolation. Indeed, in these constructs, D1 explores a range of previously-unobserved conformations that continuously connect the "partly open" (sub-state 1) to the "fully open" state (sub-state 20). Moreover, 3DVA analysis showed that the D1-3 construct is characterized by less-pronounced conformational changes with respect to D1-2, due to reduced angular motion of the 'core D1c' helix.

## D2-D4 dock sequentially and dynamically onto D1

Establishing that D2, D3, and D4 affect the D1 dynamics and induce the opening of its core to prepare for harboring the active site residues, we then sought to understand how these 3'-terminal domains dock onto D1.

In the D1-2 dataset, we can observe complete density for D2 in the "partly open" state, but not in the "fully open" state. More precisely, D2 density is visible in 3DVA sub-states 1 to 6, while in states 7-20 the θ-θ' tertiary interaction is broken and D2 becomes structurally disordered, along with 'peripheral D1c' (Fig. 5A).

Analogously, in the D1-3 dataset, we can observe density for D2 in the "partly open", but not in the "fully open" states. Moreover, we cannot observe density for D3 in either of the two states. Curiously, despite being disordered, D3 does have an impact on the D1 conformational dynamics, as observed in our 3DVA analysis (see above). More precisely, D2 density is visible in 3DVA sub-states 1 to 8, while from sub-state 9 to 20, density corresponding to $i_2$, D2, and 'peripheral D1c' gradually disappears (Fig. 5B).

Finally, in the D1-4 dataset, we can observe density for D2 and D3, but not for D4, in the "partly open" state (Fig. 3E).

In summary, these structural data suggest that D2, D3, and D4 dock one after the other onto D1, depicting the following sequence of events for the multi-domain assembly of group II introns. Steric clashes occurring between the D1 scaffold (specifically, the 'peripheral D1c' motif) and D2, the domain that is transcribed immediately after the scaffold, trigger the opening of D1. However, D2 is not stably bound to D1 in this state, but only transiently docks onto D1 by establishing the θ-θ' interaction in the 'partly open' state. Subsequently, each domain sequentially docks onto D1, but only if the preceding domains are already docked and only in the presence of the following domain. For instance, in the D1-3 state, D2 is stably docked onto D1, but not D3. D3 only docks onto D1 in the D1-4 constructs, where however, D4 remains flexible and does not establish tertiary interactions with the rest of the molecule (Figs. 3–5 and Supplementary Table 1). Surprisingly, though, this sequential multi-domain assembly process is not accomplished through the opening of the D1 pocket from the "closed" conformation progressively to the "catalytic" conformation. Instead, D1 first transits from its "closed" state into "open" states, characterized by a pocket much wider than the "catalytic" state. These "open" states are in equilibrium between two extreme conformations, the "partly" and the "fully open" states. D1 closes back into its "catalytic" state only upon D5 docking, positioning all domains in place for catalysis.

## D1 gating is mechanistically controlled by specific conformational changes of conserved hinge 1 residues

Established that D2, D3, and D4 affect D1 dynamics and sequentially dock onto this scaffold domain, we then investigated the molecular bases for the opening of D1.

To obtain detailed molecular insights into the D1 dynamics in the presence of the 3'-terminal domains, we have employed a pipeline that iteratively combines highly-selective particle exclusion and *CryoSPARC* 3D reconstruction, followed up by angular refinement. By selectively excluding low-resolution particles, we increased the proportion of high-quality particles and drastically reduced the number of particles per class, allowing us to obtain highly-resolved maps for the two herein identified "partly open" and "fully open" states of D1-2 and D1-3, respectively (Table 1).

We then reconstructed the models of the "partly open" and "fully open" states of D1-2 and D1-3 (Fig. 6). These models matched those obtained from heterogeneous refinement in *CryoSPARC* and from the 3DVA analysis, as judged by the D1 geometry ('angle A', 'gate distance', Supplementary Table 1). Importantly, the most highly-resolved maps show that, in the "partly open" conformation of both D1-2 and D1-3, hinge 1 residue A72 orients towards the core of the D1c helix, as in the "catalytic" state, but C116 bulges out of the helix, as in the "closed" state, and its solvent-exposed nucleobase remains disordered (Fig. 6A, B). In the "fully open" state of D1-3, though, both A72 and C116 orients towards the core of the D1c helix, as in the "catalytic" state (Fig. 6C, D). Hinge 2 residues (A217 and A223-A224), instead, do not show major conformational changes between the two states in either construct. Finally, the ζ and κ nucleotides remain flexible and displaced in both states of both constructs, as observed in the heterogeneous refinement and 3DVA models, and in agreement with the previously proposed induced-fit mechanism for the formation of these long-range tertiary interactions (Supplementary Fig. 13).

Next, we conducted multiple walker well-tempered metadynamics (MW-MetaD) simulations using the cryo-EM coordinates of both the "partly open" and the "fully open" states of D1-3 as starting models, to sample the conformational space and identify the lowest energy pathway connecting the cryo-EM models (Fig. 7, see also the description of the structural models used for MD simulations in the Materials and Methods section). Mechanistically, these simulations revealed that the interaction between the tertiary Z-anchor motif and

**Table 1 | Data collection, processing, and model validation table for D1-2, D1-3, and D1-4**

| State<br>PDB id<br>EMDB id | D1-2<br>partly open<br>9G4I<br>EMD-51040 | fully open<br>9G4V<br>EMD-51068 | D1-3<br>partly open<br>9G4L<br>EMD-51044 | fully open<br>9G4J<br>EMD-51041 | D1-4<br>partly open<br>9G54<br>EMD-51077 | fully open<br>9G56<br>EMD-51080 |
|---|---|---|---|---|---|---|
| **Data Collection** | | | | | | |
| Magnification | 105,000 | 105,000 | 105,000 | 105,000 | 130,000 | 130,000 |
| Voltage (kV) | 300 | 300 | 300 | 300 | 300 | 300 |
| Data collection mode | Counting | Counting | Super resolution | Super resolution | Counting | Counting |
| Electron exposure ($e-/\text{Å}^2$) | 38.07 | 38.07 | 40 | 40 | 40 | 40 |
| Defocus range (μm) | 0.8–2.2 | 0.8–2.2 | 0.8–2.2 | 0.8–2.2 | 0.8–1.8 | 0.8–1.8 |
| Pixel size (Å) | 0.839 | 0.839 | 0.42 | 0.42 | 0.81 | 0.81 |
| Symmetry imposed | C1 | C1 | C1 | C1 | C1 | C1 |
| Initial particle images (No.) | 306,972 | 306,972 | 615, 617 | 615, 617 | 349,634 | 349,634 |
| **Processing** | | | | | | |
| Name of the map | Partly open | Fully open | Partly open | Fully open | Partly open | Fully open |
| Final particle images (no.) | 41,652 | 39,075 | 18,005 | 16,493 | 41,981 | 56,717 |
| Map resolution (Å) | 4.61 | 4.69 | 4.01 | 3.74 | 6.30 | 7.54 |
| FSC threshold | 0.143 | 0.143 | 0.143 | 0.143 | 0.143 | 0.143 |
| Map local resolution range (Å) (FSC Threshold 0.5) | 4.0-6.7 | 4.1-6.8 | 3.6-9.5 | 3.5-9.2 | 7.0-10.0 | 7.0-10.0 |
| **Model Refinement** | | | | | | |
| Geometry refinement | ERRASER | ERRASER | ERRASER | ERRASER | - | - |
| Real space refinement | TEMPy-ReFF | TEMPy-ReFF | TEMPy-ReFF | TEMPy-ReFF | TEMPy-ReFF | TEMPy-ReFF |
| Initial model used (PDB id) | 4FAQ | 4FAQ | 4FAQ | 4FAQ | 4FAQ | 4FAQ |
| Model resolution (Å) | 4.6 | 4.7 | 4 | 3.7 | 6.5 | 7.5 |
| FSC threshold | 0.143 | 0.143 | 0.143 | 0.143 | 0.143 | 0.143 |
| **Model composition** | | | | | | |
| Total atoms | 5845 | 4715 | 5714 | 5157 | 6288 | 4631 |
| Hydrogen atoms | 0 | 0 | 0 | 0 | 0 | 0 |
| Non-hydrogen atoms | 5845 | 4715 | 5714 | 5157 | 6288 | 4631 |
| Nucleotide | 273 | 221 | 267 | 241 | 293 | 217 |
| **R.M.S. deviations / (num. outliers)** | | | | | | |
| Bond lengths (Å) | 0.014(0) | 0.014(0) | 0.014(0) | 0.014(0) | 0.014(0) | 0.014(0) |
| Bond angles (°) | 1.72 (29) | 153 (8) | 1.48 (9) | 1.47 (8) | 1.78 (57) | 1.75 (27) |
| **Validation** | | | | | | |
| Validation software | Phenix | Phenix | Phenix | Phenix | Phenix | Phenix |
| MolProbity score | 1.77 | 1.65 | 1.65 | 1.65 | 1.69 | 1.71 |
| Clashscore | 0.34 | 0.00 | 0.00 | 0.00 | 0.11 | 0.14 |
| $CC_{mask}$ | 0.85 | 0.85 | 0.86 | 0.86 | 0.90 | 0.86 |
| $CC_{peak}$ | 0.79 | 0.73 | 0.79 | 0.77 | 0.82 | 0.70 |
| $CC_{volume}$ | 0.84 | 0.84 | 0.86 | 0.86 | 085 | 0.79 |
| Average Suitness | 0.508 | 0.553 | 0.666 | 0.669 | 0.490 | 0.441 |
| Validation software | TEMPy | TEMPy | TEMPy | TEMPy | TEMPy | TEMPy |
| SMOC Score | 0.90 | 0.90 | 0.87 | 0.83 | 0.91 | 0.92 |

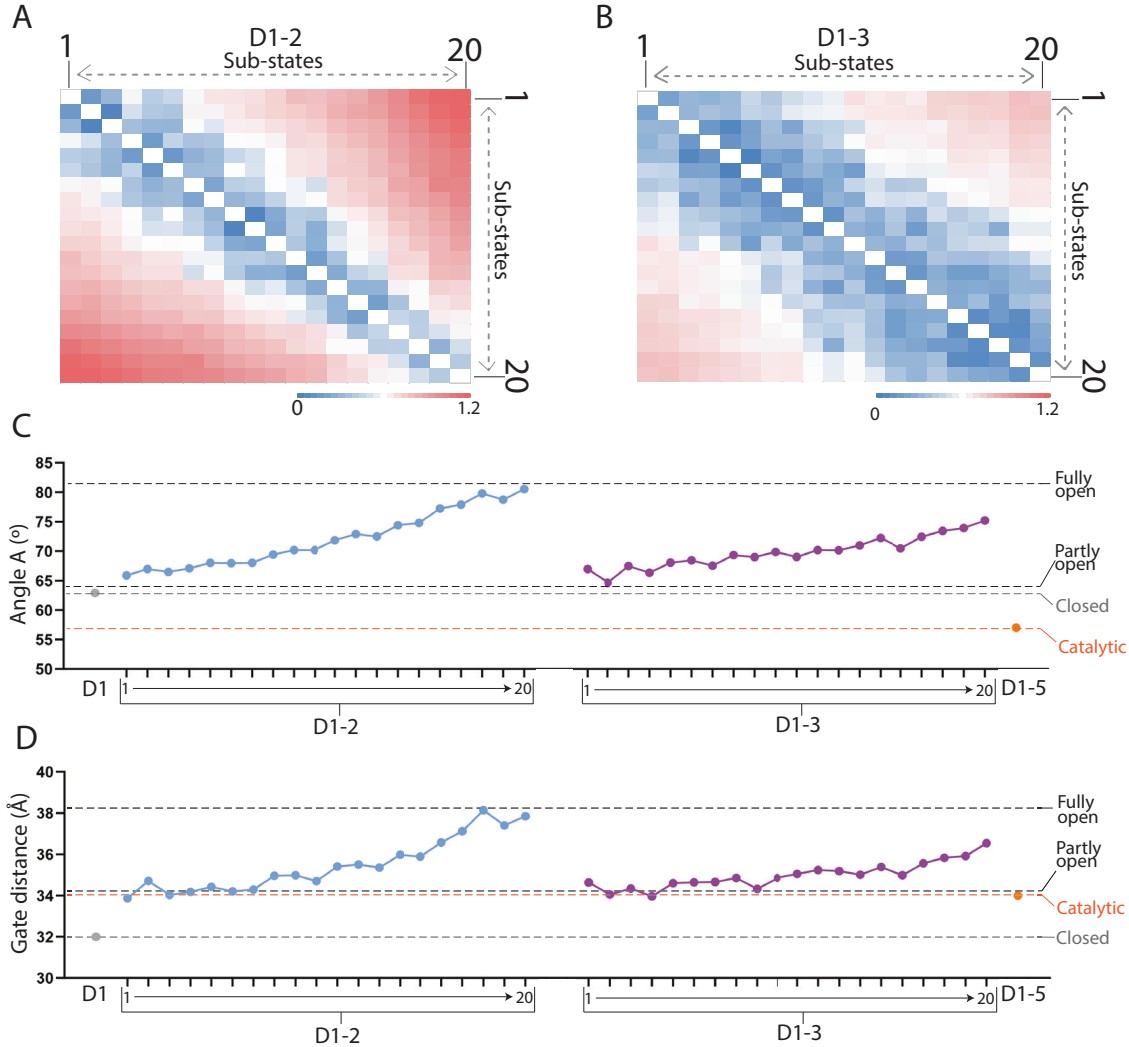

**Fig. 4 | Quantification of D1 dynamics. A** Pairwise RMSD heatmap of models generated for the sub-states 1-20 obtained from 3DVA of D1-2. **B** Pairwise RMSD heatmap of models generated for the sub-states 1-20 obtained from 3D variability analysis of D1-3. **C** Scatter line plot of 'angle A' obtained for isolated D1 ("closed", grey), sub-states 1–20 of D1-2 (blue), sub-states 1–20 of D1-3 (purple), and D1-5 ("catalytic", orange). The y-axis displays 'angle A' (in degrees). **D** Scatter line plot of 'gate distance' obtained for D1 ("closed" state, grey), sub-states 1-20 of D1-2 (blue), sub-states 1-20 of D1-3 (purple), and D1-5 ("catalytic", orange). The y-axis represents the 'gate distance' (in Å). Source data are provided as a Source Data file.

the residues in the $i_1$-$i_2$ segment appears to play a key role for the transition. In the "partly open" conformation, the tertiary Z-anchor motif is stabilized by stacking interactions between A12 and A110 and between G108 and C262, respectively. Hydrogen bond interactions are formed and maintained between A110 and U259 and between C11 and G108, respectively (Fig. 7C). For the transition from "partly open" to "fully open" to occur, this network of molecular interactions needs to change to generate an "intermediate" state. In this "intermediate" state, residue U13 plays a crucial role by replacing residue A12, forming a stacking interaction with A110 (Fig. 7C). During this transition, the Z-anchor nucleotides makes fewer interactions with the $i_1$-$i_2$ segment, but more local interactions with other D1c residues. Specifically, within the Z-anchor motif, residues G108 and A110 now form stacking interactions. Finally, in the "fully open" structure, the stacking interaction between U13 and A110 is maintained. In summary, despite rearranging intramolecular interactions, the energetic cost for D1 to transit between "partly open" and "fully open" conformations is minimal, explaining the continuum equilibrium of "open" states that we experimentally captured by cryo-EM.

More quantitatively, from our free energy surface (FES) estimation, the "partly open" state is the deepest energy minimum,

-2 kcal•mol⁻¹ lower in energy than the "fully open" state, passing through a metastable "intermediate" state, ~4 kcal•mol⁻¹ higher than the "partly open". Notably, conformational transitions between such minima are separated by activation barriers of 6–8 kcal•mol⁻¹ from "partly open" to "intermediate" and 8–10 kcal•mol⁻¹ from "fully open" to "intermediate", similar in magnitude to the transitions observed in the D1 and D1-2 simulations (see above and Fig. 7A). The fact that these energy barriers are nearly equivalent explains why all minima are observed during our enhanced sampling simulations. These barrier heights are consistent with our observation of a continuum of intermediate conformations, including the "partly open" and "fully open" principal states, observed during cryo-EM cooling[38], as identified by the 3DVA analysis.

In summary, the energy landscape features distinct basins that are connected by accessible pathways, enabling D1 to sample a wide range of dynamic "open" conformations when D2, D3, and D4 are present. These results point to a finely orchestrated conformational change of key conserved residues in and around hinge 1, which directly correlates with the orientation of the 'core D1c', 'peripheral D1c' and $i_1$-$i_2$ helices, and thus with the size and shape of the D1 pocket that has to prepare for accommodating the active site residues.

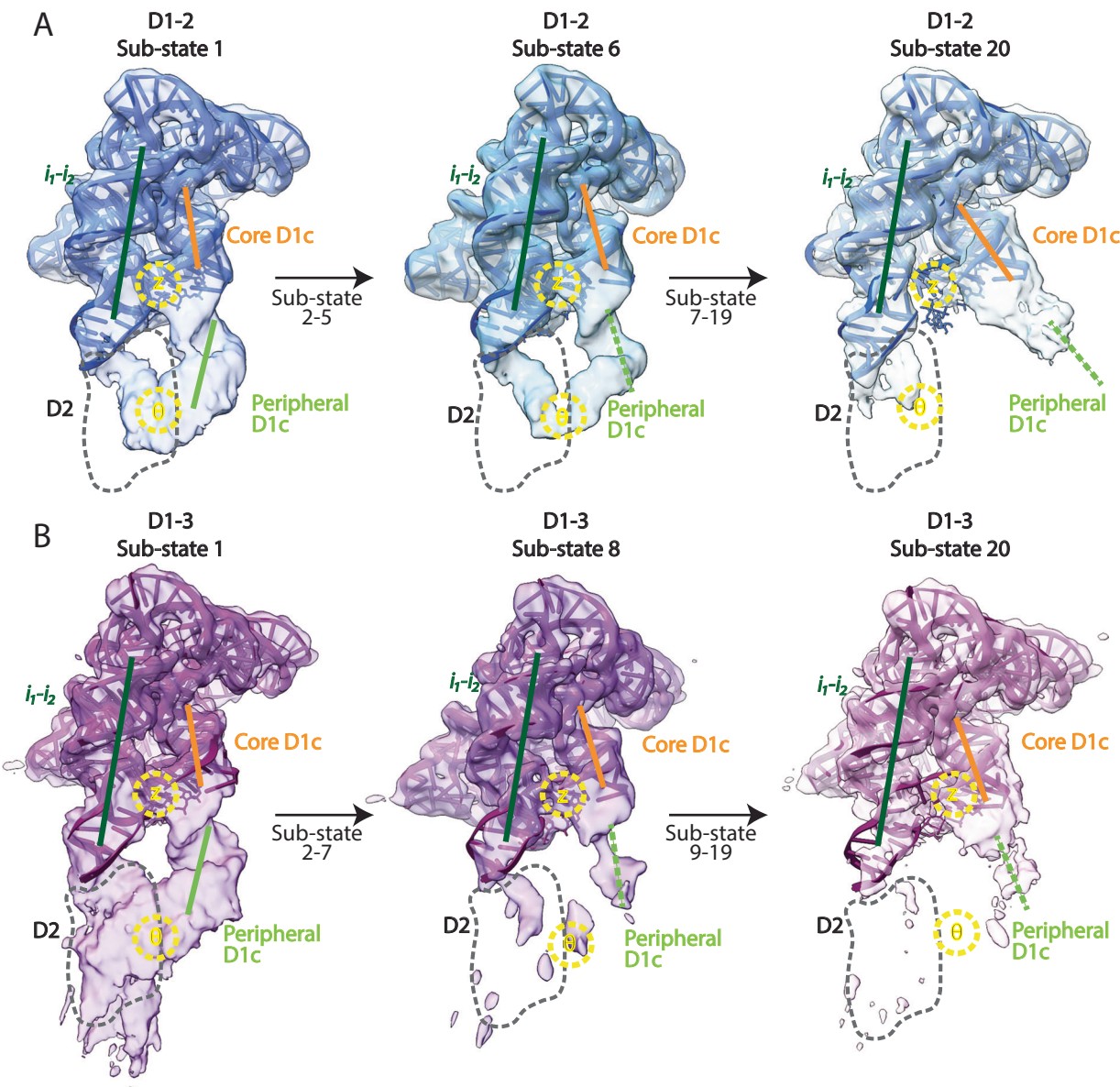

**Fig. 5 | Dynamics of D1 in the presence of D2 and D2-3. A** Density map and model of sub-state 1 (left), 6 (middle), and 20 (right) from 3DVA of D1-2. **B** Density map and model of sub-state 1 (left), 8 (middle), and 20 (right) from 3DVA of D1-3. In all panels, thick lines represent the axes of 'core D1c' (orange), 'peripheral D1c' (light green) and $i_1$-$i_2$ (dark green). The expected position of D2 and D3 is indicated with a dark grey dashed circle. Location of tertiary motif Z-anchor and θ is depicted with yellow labels and dashed circles.

## Reducing hinge 1 dynamics by structure-based mutagenesis impairs catalysis

Establishing how hinge 1 residues mechanistically control the D1 dynamics, and thus the assembly of the group II intron structure into its functionally active conformation, we set out to determine if hinge 1 mutations have an impact on catalysis. In the past, hinge 1 had already been identified as a control hub for group II intron compaction, but in the absence of structural data, mutations designed to change the sequence of this motif could not identify functional defects[39]. Here, we revisited those earlier studies in light of our structural data and specifically hypothesized that the structural dynamics of hinge 1, rather than the sequence identity of hinge 1 residues, are important to control group II intron assembly (Fig. 8).

First, we mutated the *O. iheyensis* I1 intron to reduce the dynamics of hinge 1 by locking into Watson-Crick base pairs the 5′- and the 3′-sides of this internal loop (residues 71–73 and 114–116, respectively). Splicing kinetics reveal that the mutant performs the first step of splicing 6-fold slower than the wild type ($k_{1\text{-pOiA}} = 0.016 \pm 0.003$ min$^{-1}$; $k_{1\text{-pOiA-hinge1mut}} = 0.003 \pm 0.000$ min$^{-1}$) and the second step of splicing 2-fold slower than the wild type ($k_{2\text{-pOiA}} = 0.041 \pm 0.007$ min$^{-1}$; $k_{2\text{-pOiA-hinge1mut}} = 0.028 \pm 0.005$ min$^{-1}$, Fig. 8A, B and Supplementary Table 4).

Second, to probe the evolutionary sequence, structural and functional conservation of hinge 1, we introduced homologous mutations also in the yeast *ai5γ* group II intron. We detected equally severe functional defects. Specifically, the mutant (ai5γ-D135-hinge1-mut) has a $v_{max} = 0.75 \pm 0.06$ nmol•L$^{-1}$•min$^{-1}$, corresponding to 3 % of the $v_{max}$ of wild-type (ai5γ-D135, $v_{max} = 22.47 \pm 9.49$ nmol•L$^{-1}$•min$^{-1}$), and a $K_M = 7.87 \pm 3.24$ nmol•L$^{-1}$, corresponding to 4 % of the $K_M$ of wild-type ($K_M = 192.67 \pm 80.93$ nmol•L$^{-1}$, Fig. 8D, E and Supplementary Table 4).

In summary, our systematic structural characterization of group II introns assembly enabled us to identify specific, conserved residues that allosterically control catalysis by regulating the folding dynamics

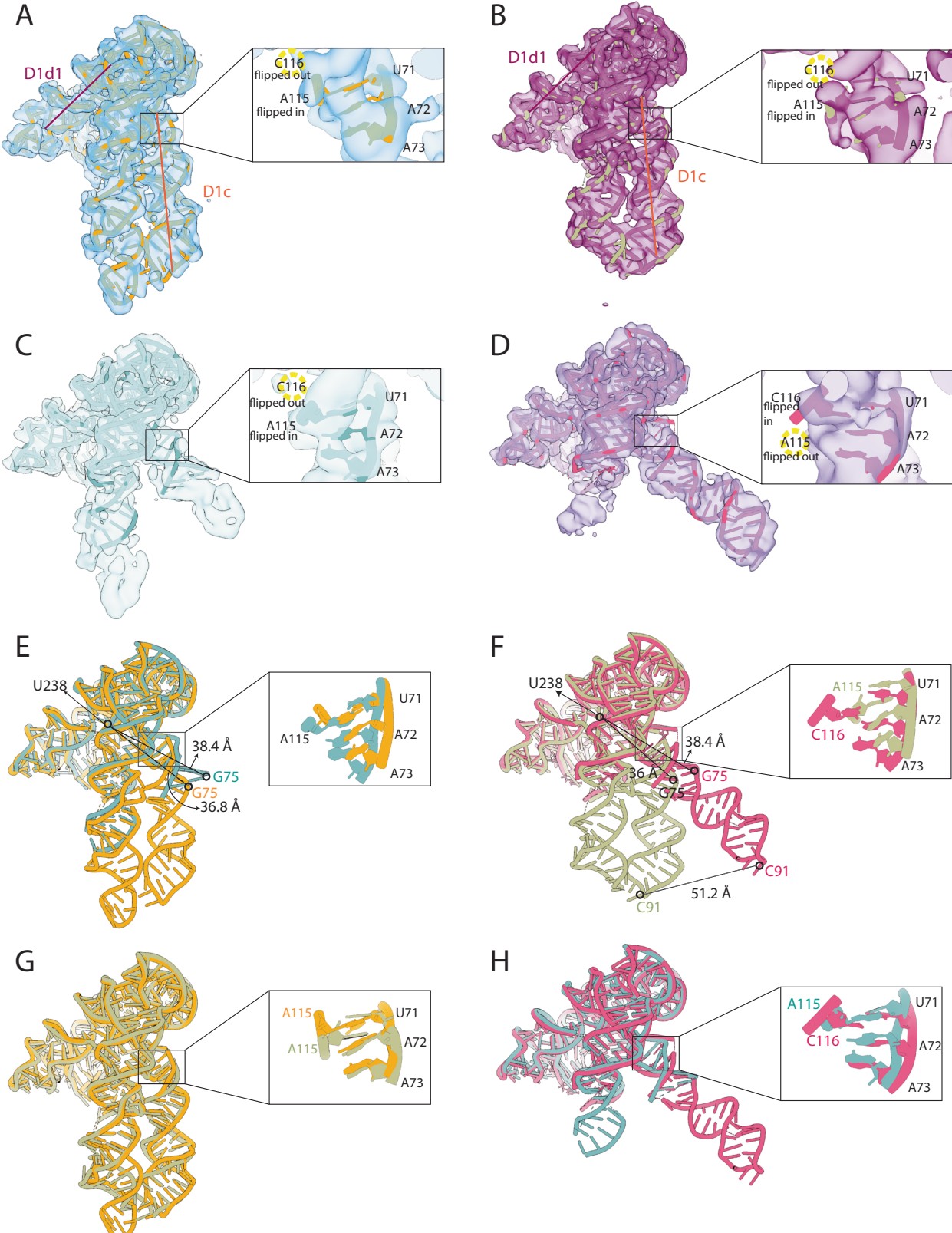

**Fig. 6 | Dynamics of hinge 1 residues in D1-2 and D1-3. A** Sharpened density maps and models of the "partly open" state of D1-2. **B** Sharpened density maps and models of the "partly open" state of D1-3. **C** Sharpened density maps and models of the "fully open" state of D1-2. **D** Sharpened density maps and models of the "fully open" state of D1-3. In **A–D**, the black square highlights the region of hinge 1 nucleotides, which is magnified in the zoomed-in insets. The position and conformation (flipped in *vs* flipped out) of residues A115 and C116 is indicated by dashed yellow circles. **E** Superposition of "partly open" (orange) and "fully open" (teal) states of D1-2. The *gate distance* in the two states is indicated as a black dotted line. **F** Superposition of "partly open" (light green) and "fully open" (pink) states of D1-3. The *gate distance* in the two states is indicated as a black dotted line. The distance between C91 in the "partly open" and C91 in the "fully open" states is indicated as a grey dotted line. **G** Superposition of "partly open" state of D1-2 (orange) and "partly open" state of D1-3 (green). **H** Superposition of "fully open" state of D1-2 (light blue) and "fully open" state of D1-3 (purple).

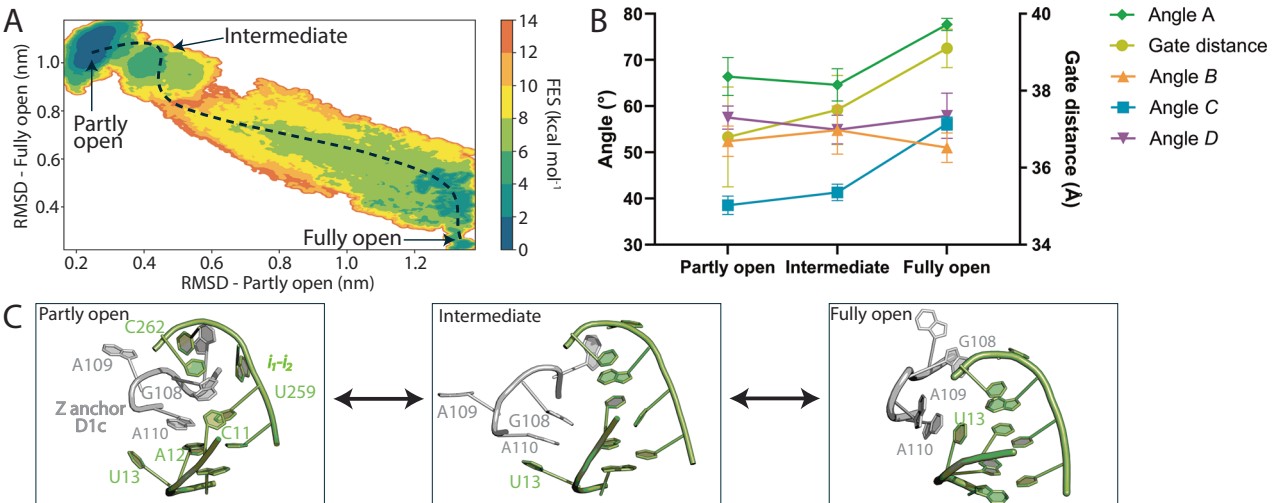

**Fig. 7 | Molecular dynamics. A** Free energy profile from multiple walker well-tempered metadynamics simulations illustrated as a function of the RMSD (in nm) with respect to the "partly open" (x-axis) and "fully open" (y-axis) states of the D1-3 system. **B** Geometric descriptors of the free energy minima. '*Angles A-D*' and '*gate distance*' are defined in Fig. 1. The error bars represent standard deviation (S.D.) obtained from 10 frames for each energy minimum. Source data are provided as a Source Data file. **C** Cartoon representation of the molecular structure of the Z-anchor (grey) and $i_1$-$i_2$ (green) for the "partly open", "intermediate", and "fully open" states.

of this ribozyme. The second step splicing defect of the *O. iheyensis* intron mutant further suggests that these dynamics also control previously described conformational rearrangements that need to occur during catalysis, between the first and the second steps of splicing[23,30].

## Discussion

In this work, we have visualized the multidomain assembly pathway of a large non-coding RNA of biomedical and bioengineering significance, the group II self-splicing intron, in the process of forming a catalytically active tertiary structure. The distinctive property of our target is its ability to position peripheral domains onto a pre-assembled structural scaffold without forming inactive, misfolded states ('kinetic traps'), which are instead commonly generated by many naturally-occurring RNAs[6]. Understanding the molecular mechanism by which large RNAs avoid 'kinetic traps' is important to improve RNA engineering and enhance the design of more efficient RNA molecular machines.

Setting a distinctive mark with respect to recent studies that visualized static snapshots of kinetically-trapped RNAs[9,17–19], our cryo-EM study now captured the dynamic assembly of a large RNA into its functional conformation. We present an ensemble of 3D structures of group II intron intermediates, corroborated and mechanistically enriched by in-solution SAXS data and extended MD simulations and free energy calculations, which reflect the complex dynamic conformational space explored by our target. Our structures offer molecular insights and a precise quantitative description of the structural dynamics of the group II intron scaffold, formed by domain D1, and how such dynamics facilitate the assembly of the catalytically-active ribozyme. We define this assembly mechanism as the 'dynamic gate mechanism' to highlight the crucial role of D1 dynamics in regulating the docking of the active site motifs into its core (Fig. 9 and Supplementary Movie 1).

Our data show that such a 'dynamic gate mechanism' relies on two fundamental and distinctive principles, which seem strategic for all RNAs that fold onto a pre-assembled scaffold, avoiding 'kinetic traps', and may thus represent general RNA folding paradigms.

The first principle consists of the fact that the assembly of peripheral domains onto a pre-assembled scaffold needs to be precisely organized and hierarchically regulated. In the group II intron, our structural data show that D2, D3, and D4 dock one after the other onto D1, in a sequential manner from the 5'- to the 3'-end of the molecule, according to the previously-proposed "first come, first fold" mechanism[28] (Figs. 3–5 and Supplementary Table 1). This strategy likely ensures that peripheral domains do not cause misfolding of the scaffold. In group II introns, this strategy is specifically needed to avoid D4 interfering with the folding of D1, because D4 harbors a long open reading frame encoding for the intron-associated protein maturase[21]. Only in the last assembly step, when D5 enters the "open" core of D1, the molecule locks into a single "catalytic" conformation capable of catalysis, as previously captured crystallographically[29].

Unexpectedly, though, our work also shows that such a sequential multi-domain assembly process is not accomplished through the progressive opening of the D1 pocket from the "closed" to the "catalytic" state. Instead, D1 first transits from its "closed" state into "open" states, characterized by a pocket much wider than the "catalytic" state. This observation defines the second distinctive principle for RNAs to avoid 'kinetic traps'. The pre-assembled scaffold needs to be dynamic and able to sample a wide range of conformations separated by low energy barriers. In our work, we have specifically captured a continuum of previously-unrecognized D1 "open" conformations – in equilibrium between two extreme and energetically slightly favored states, the "partly open" and the "fully open" states. These "open" states are sampled by D1 only in the presence of 3'-terminal domains (D2-D4), and differ from the conformations of both D1 in isolation ("closed" state), and of D1 in the functionally active ribozyme ("catalytic" state). These "open" D1 conformations are absolutely required for group II intron assembly, because they widen the D1 pocket largely to receive the active site residues. As such, RNA follows the same principles of a nutcracker. Intuitively, one would open a nutcracker widely to comfortably insert the nuts before cracking them. Similarly, the group II intron scaffold opens widely to recruit the active site and splice junctions in its core, before splicing. A rigid scaffold that explores only conformations similar to the fully folded molecule would be much less suited to guide the folding process and would increase the risk of forming unproductive interactions and kinetically trapped conformations. In group II introns, the dynamics displayed by the "open" conformations, toggling between "partly" and "fully" open states, is likely beneficial to facilitate the efficient formation of long-range tertiary contacts, such as θ-θ' (between D1c and D2), and λ-λ' and κ-κ' (between D1c and D5), which form by induced fit and lock-and-key

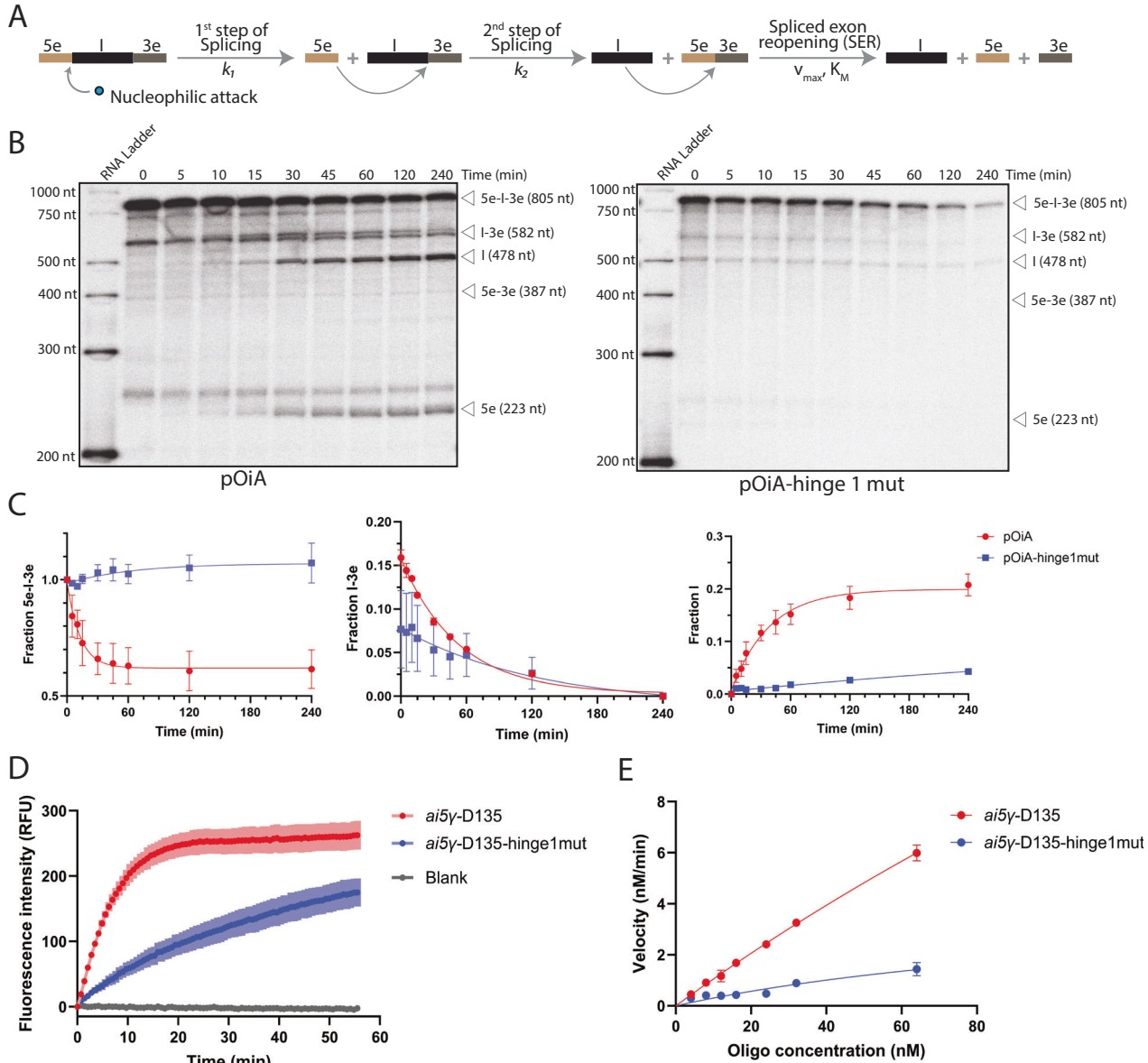

**Fig. 8 | Functional assays. A** Schematic representation of the group II intron splicing reaction. $k_1$ is the rate constant of the first step and $k_2$ of the second step of splicing; $v_{max}$ is the maximum velocity, and $K_M$ is the substrate concentration at which the reaction rate is half of its maximum velocity. **B** Representative splicing kinetics of pOiA and pOiA-hinge1mut. The size of the precursor and products is indicated on the right side of each gel. 'Ladder' is an in vitro transcribed and radiolabeled RNA molecular weight size marker (size in nucleotide on the left side of each gel). **C** Evolution of the populations of pOiA and pOiA-hinge1mut precursor (5e-I-3e, left panel), intermediate (I-3e, middle panel), and linear intron (I, right panel) over time. Error bars represent standard errors of the mean (s.e.m.)

calculated from $n = 3$ independent experiments. $k_1$ and $k_2$ are indicated in Supplementary Table 4. **D** Evolution of the population of splice exons generated by ai5γ-D135 and ai5γ-D135-hinge1mut in a spliced-exon reopening (SER) assay using a FRET-labeled oligo representing the ligated exons as substrate (representative curve at 64 nM oligo final concentration). Error bars represent standard errors of the mean (s.e.m.) calculated from $n = 3$ independent experiments. **E** Michaelis-Menten plot reporting the kinetic properties of ai5γ-D135 and ai5γ-D135-hinge1mut ($v_{max}$ and $K_M$ values are indicated in Supplementary Table 4). Error bars represent standard errors of the mean (s.e.m.) calculated from $n = 3$ independent experiments. Source data are provided as a Source Data file.

mechanisms, respectively[21]. These observations help explain a conundrum in group II intron folding, i.e., the combination of their slow folding rate (minutes[26]) and the lack of kinetic traps[26,40]. Possibly, establishing a conformational equilibrium between "open" states is a useful strategy that group II introns adopt to control the slow formation of long-range tertiary contacts between regions located far apart from each other in the secondary structure[26]. The small but not negligible energy barriers that we capture by MD simulations may be functional to coordinate these steps. With respect to long-range interactions, our results specifically explain at the molecular level many biochemical experiments previously reported on group II intron

folding[8,41]. For instance, those earlier studies had observed by hydroxyl radical footprinting and dimethyl sulfide probing that the θ-θ' interaction between D1c and D2 is only stably established in the presence of D5. In line with these chemical probing data, here we show that θ-θ' is not stably formed in the "open" conformations of D1-2, D1-3, and D1-4. Moreover, our structures reveal that the tertiary motifs κ, ζ, and λ also remain flexible throughout the group II intron assembly process, in line with other previously reported chemical probing experiments, which had established that these motifs are chemically reactive under near-physiological conditions, until D5 has docked onto D1 to form the catalytically-active intron[27,42]. Thus, our data confirm the existence and

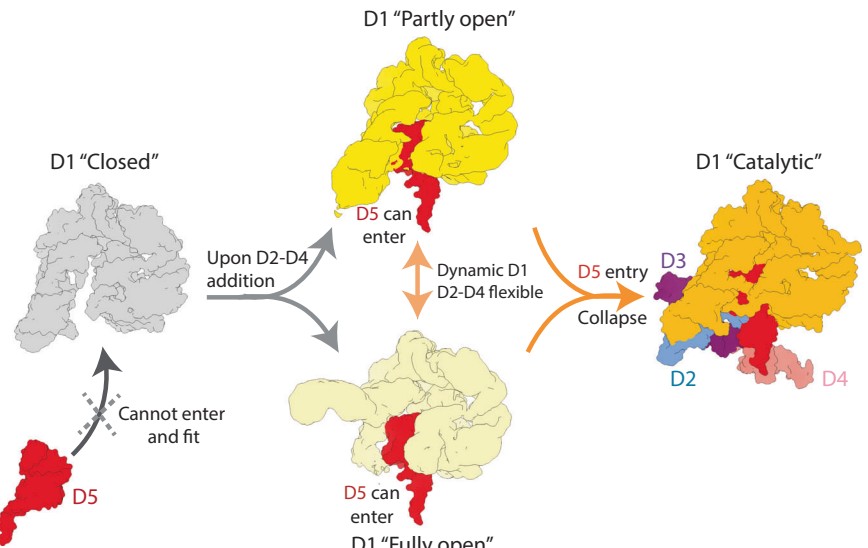

**Fig. 9 | Mechanistic model for group II intron assembly.** D1 transitions from a "closed" state (grey) to a continuum of "open" states in equilibrium between two extreme conformations, i.e., the "partly open" (dark wheat) and "fully open" (light wheat) states. The middle arrow in wheat color represents the continuous dynamics of the D1 between "partly open" and "fully open" states. Upon D5 docking (red), D1 closes back into its "catalytic" state (orange).

provide a direct visualization via cryo-EM of dynamic group II intron assembly intermediates postulated two decades ago.

It has to be said that not observing 'kinetic traps' does not necessarily exclude their formation. As a matter of fact, during grid optimization, we observed particle unfolding, aggregation clusters, and non-specific ice behaviors, e.g., leopard ice, which we minimized by optimizing RNA concentration, blot force, and blot time at the plunge freezing step. We also optimized each data set by manually curating the micrographs, excluding unfolded and aggregated particles and retaining the high-quality images most suitable for particle picking, 2D classification, and 3D reconstruction (see Materials and Methods for more details). But crucially, to determine the structure of all constructs, we maintained close-to-neutral pH and close-to-intracellular concentrations of physiological ions, and in general, we used biochemical conditions that are optimal for splicing. The fact that our structure-based design of hinge 1 mutations led to the production of a catalytically inefficient construct suggests that our structural work captures functionally meaningful molecular mechanisms.

From our data, the hinge 1 motif emerges as a crucial determinant for the allosteric modulation of the splicing reaction, as also suggested by recent studies that structurally characterized the first step of the splicing reaction *via* the branching pathway[43]. Within hinge 1, residues A72 and C116 specifically stand out as the pivot point that regulates the opening and closing of the D1 'dynamic gate'. These residues transition from a bulged-out conformation characteristic of the "closed" state (the isolated D1 scaffold), to a hybrid bulged-in/bulged-out conformation characteristic of the "open" states (D1 in the presence of D2 and D3), to a fully bulged-in conformation characteristic of the "catalytic" state. Notably, hinge 1 residues are conserved and adopt an internal loop structure across all classes of group II introns, from bacteria to yeast and plant organelles, suggesting that the conformation of the D1c stem and its internal loop separating 'core D1c' from 'peripheral D1c' is important. Interestingly, in many introns, the hinge 1 motif contains insertions that form additional long-range interactions (Supplementary Fig. 14). For instance, long helical insertions at positions homologous to C116 in group IIA and IIB introns establish kissing loop interactions (β-β') within D1. Instead, short single-nucleotide insertions in the group IIC intron from *Eubacterium rectalis* form interactions with D5[43]. These evolutionary considerations – along with the functional defects that

we observed for hinge 1 mutants in homologous bacterial and yeast introns – point to the fact that the hinge 1 motif is a critical functional element for regulating access to the D1 core and for positioning peripheral D1 nucleotides in the correct orientation to form stabilizing long-range interactions needed for catalysis.

The hinge 1 motif, and particularly the toggling nucleotides A72 and C116, thus emerges as a potentially vulnerable site, which could be exploited to trap group II introns in unproductive conformations, for instance, using small molecule inhibitors. Complementary to direct targeting of the intron active site[23], such RNA-directed drug design strategy could be of important relevance for human health, because the self-splicing group II introns are validated therapeutic targets to treat fungal infections[44].

Besides highlighting the complex D1 dynamics, our group II intron assembly study also provides specific mechanistic insights of potentially broader relevance for other RNAs. The dynamic interplay between RNA helices, which is a characteristic feature of our 'dynamic gate mechanism', represents a more-generalizable fundamental principle to guide RNA assembly across biological systems, and ensure that functional RNA components are assembled precisely, in a tightly-controlled fashion. For instance, in riboswitches – which represent another class of regulatory non-coding RNAs – the assembly and dynamics of helices in the aptamer domain dictate the specific conformation adopted by the RNA and decide the transcription fate. This concept is well exemplified by the manganese-sensing transcriptional riboswitches, whose aptamer subdomain helices acquire undocked, semi-docked, or docked conformations in different assembly states[45]. Structurally and functionally important angular movements of RNA helices also occur in larger RNAs, like the ribosome. For example, during processome assembly, helical rRNA subdomains undergo substantial angular motion to acquire their pre-40S conformation[46]. Analogously, during co-transcriptional translation, helix H44 in the bacterial rRNA undergoes a pronounced angular motion to facilitate the entry of newly transcribed mRNA emerging from the RNA polymerase[47].

Finally, besides elucidating key aspects of RNA assembly and evolution, our data also showcase the power of cryo-EM to resolve the complex dynamic behavior of large RNA molecules, based on established 3D classification approaches. These approaches remain essential, as RNA structure prediction algorithms, including AI-based ones

like AlphaFold3, may not always reflect the dynamic behavior of RNA[48–50]. By employing manually curated 3D classification, we achieved robust and reliable structural interpretations while minimizing the likelihood of hallucinations associated with self-supervised learning frameworks, such as variational autoencoders employed in cryo-EM data processing methods like CryoDRGN, particularly under conditions of low signal-to-noise ratios[37,51,52]. Through our work, we demonstrate the feasibility of using and acquiring biologically relevant information from 3D variability analysis for *protein-free* RNA samples.

In summary, by integrating the latest advances in single particle cryo-EM and image processing with enzymatic, biochemical, biophysical and MD simulation studies to determine the assembly mechanism and resolve the complex structural dynamics of a group II self-splicing intron, our work provides a near-atomic resolution molecular movie of a large multidomain RNA in the process of acquiring its functionally active conformation and establishes a basis for understanding how RNA uses the conformational dynamics of its helical and junction motifs to avoid the formation of non-functional 'kinetic traps' (Supplementary Movie 1).

## Methods

### Cloning
The D1, D1-5, and pOiA constructs of the *O. iheyensis* group II intron and the D135 construct of the *S. cerevisiae* ai5γ group II intron (ai5γ-D135) have been described previously[28,29,39]. Here, we have generated the *O. iheyensis* group II intron D1-2 and D1-3 constructs from the D1 vector using sequence- and ligation-independent cloning (SLIC) to add domains D2 and D2-3 before the BamHI site, respectively. Moreover, we have generated the *O. iheyensis* group II intron D1-4 construct from the D1-5 vector by SLIC by removing D5. We note that we have used a group II intron crystallization construct to design the constructs of the domain assembly intermediate. In the crystallization construct, D2, D3, and D4 have truncated sequences and not the wild-type sequence. The truncated regions are not required to stabilize the D1 in "catalytic" state, as it can be observed from the previous crystal structure of group II intron (PDB id: 4FAQ). Finally, we have generated the *O. iheyensis* group II intron hinge 1 mutant (pOiA-hinge1mut) from pOiA by mutating residues $U^{71}A^{72}A^{73}$ to GUC to base pair with complementary nucleotides $G^{114}A^{115}C^{116}$ and lock hinge 1; and we have generated the *S. cerevisiae* ai5γ group II intron hinge 1 mutant (ai5γ-D135-hinge1mut) from ai5γ-D135 by exchanging residues $G^{124}A^{125}C^{126}$ to a U, to base pair with $A^{77}$ and lock hinge 1. We validated the sequence of all vectors by DNA sequencing (Eurofins). We used *E. coli* Mach1 competent cells for cloning and plasmid amplification. We extracted plasmids with mini and maxi prep kits (QIAGEN) from single colonies. All primers used for cloning are listed in Supplementary Table 5.

### In vitro transcription and purification
We produced and purified D1, D1-2, D1-3, D1-4, pOiA, pOiA-hinge1mut, ai5γ-D135, and ai5γ-D135-hinge1mut by non-denaturing methods[34]. Briefly, we linearized all vectors using the BamHI restriction enzyme (NEB). Then, we used linearized vectors for in vitro transcription with T7 polymerase in the buffer containing 22 mM $MgCl_2$, 40 mM Tris-HCl, pH 8.0, 0.2 mM spermidine, 10 mM DTT, and 0.01% Triton X-100 100%. Following transcription, we sequentially removed template DNA and proteins using Turbo DNase (Thermo Scientific) and proteinase K (Promega), respectively. Finally, we rebuffered RNA into 10 mM $MgCl_2$, 5 mM Na-MES pH 6.5 buffer using Amicon Ultra-0.5 centrifugal concentrators (molecular weight cut-off of 100 kDa).

### Gel electrophoresis
We performed gel electrophoresis using 2% agarose gels containing 5 μL of SYBR Safe dye. Before loading the RNA sample in the gel, we diluted the sample 10 times with rebuffering buffer to a final volume of 10 μL. Then, we added 2 μL of 6x RNA native gel dye (0.5x TB buffer, 40% sucrose, 0.5% w/V Orange G). We ran all the gels in 1x TB buffer (89 mM Tris base, 89 mM boric acid) at 110 V for 30 min. After electrophoresis, we imaged the gels on a Gel Doc™ XR+ (Bio-Rad).

### Size-exclusion chromatography (SEC)
We used purified RNA at concentrations of 10 μM to inject on a SEC column using Tricorn columns (GE Healthcare) self-packed with Sephacryl S400 resin. We pre-equilibrated the column in 10 mM $MgCl_2$ and 5 mM Na-MES pH 6.5 buffer and performed the chromatographic run in the same buffer.

### Size-exclusion chromatography coupled to small-angle X-ray scattering (SEC-SAXS)
We performed size exclusion chromatography coupled to small-angle X-ray scattering (SEC-SAXS)[53,54] using centrifugal filters (pore size 0.22 mm) to filter the RNA before injection. We injected 10 μM RNA on Tricorn columns (GE Healthcare) self-packed with Sephacryl S400 resin. We used 50 mM $MgCl_2$, 150 mM KCl, and 5 mM Na-MES pH 6.5 as running buffer. We acquired SEC-SAXS data at the BioSAXS beamline BM29 at ESRF, Grenoble. We analyzed the data using ATSAS modules CHROMIXS, PRIMUS, CRYOSOL, GAJOE, and DAMMIF[55]. We used CHROMIXS to visualize the SEC-SAXS output of scattering intensity and UV absorbance. We selected suitable frames for buffer subtraction and suitable frames corresponding to the elution of the sample to obtain the scattering curves. We used PRIMUS to visualize the scattering curves and obtain hydrodynamic parameters, such as the radius of gyration ($R_g$) and maximum pairwise distance ($D_{max}$). We used CRYOSOL to compare the available D1 crystal structures with the scattering intensity curves. We used GAJOE to compare the "partly open" and "fully open" ensembles of D1 obtained from 3DVA analysis with the scattering curves.

### Mass photometry (MP)
We carried out MP experiment[56,57] by thoroughly cleaning coverslips using MilliQ water and isopropanol. Then, we coated the coverslip with 0.01 % poly-L-lysine (Sigma-Aldrich). We used 2–20 nM concentrations of RNAs for our measurements. We used the RNA Century™ Markers ladder (Applied Biosystems) to generate standard curves. We acquired MP data using the Acquire MP software and the Acquire Measure software to determine the molecular weight of the samples.

### Splicing assays
We in vitro transcribed radiolabeled intron precursors (pOiA and pOiA-hinge1mut) in the presence of α-$^{32}$P UTP (Revvity), purified them by gel electrophoresis, refolded them by denaturation at 95 °C for 1 min in the presence of 40 mM Na-MOPS pH 7.5, and cooled them at room temperature for 2 min[29]. Then, we added KCl to a final concentration of 150 mM and $MgCl_2$ to a final concentration of 5 mM[29]. We took aliquots at the indicated time points, quenched them with urea, and analyzed them on a 5% denaturing polyacrylamide gel[29]. We quantified the gels in QuantityOne (v 4.6.6, Biorad) and calculated the kinetic rate constants ($k_1$, $k_2$) of wild type and mutated intron using Kintek Explorer (v 8.0.190104, Kintek Global). For representation purposes, we plotted the data using the Prism8 package (GraphPad Software)[29]. We run all experiments in triplicate. We report the data as average ± standard error of the mean (s.e.m.).

### Fluorescence resonance energy transfer (FRET)-based spliced exon reopening (SER) assay
We performed the FRET-based SER assay[23] using refolded purified ai5γ-D135 and ai5γ-D135-hinge1mut by denaturation at 95 °C for 2 min in the presence of 50 mM K-MOPS pH 7.5, and then cooled them to

room temperature for 2 min. We added KCl to a final concentration of 150 mM to reach a concentration of 200 nM intron. We prepared the fluorescently labeled RNA oligo substrate [Integrated DNA Technologies, CGUGGUGGGACAUUU/iAlex555N/CGA/3BHQ_2/[23]] as 10x concentrated stocks at concentrations of 0, 40, 80, 120, 160, 240, 320, and 640 nM in reaction buffer consisting of 50 mM K-MOPS pH 7.5, 150 mM KCl and 300 mM MgCl$_2$. We prepared 96-well PCR plates (Azenta Life Sciences) by pipetting 9 µL of a mixture of the purified and refolded intron plus 10% DMSO at the bottom of each well, and 1 µL of 10x oligo solution on the side of each well. We then spun the plate immediately before the run in a vertical table-top centrifuge to mix the intron and oligo solutions and reach a final concentration of 160 nM intron, 50 mM K-MOPS pH 7.5, 150 mM KCl, 30 mM MgCl$_2$, and 0, 4, 8, 12, 16, 24, 32, and 64 nM oligo, respectively. We incubated the 96-well plate at 37 °C in an RT-PCR machine (Biorad), measuring the HEX channel fluorescence of each well every 30 s. We measured each condition in technical triplicate in the same plate. Additionally, we performed three independent replicas for each plate (a total of nine measurements per data point). We normalized the data by subtracting the averaged blank signal (no oligo). We calculated the $v_{max}$ and $K_M$ values by measuring the initial slope (first 10 cycles) of the response curves and by fitting the data using Michaelis-Menten and Lineweaver-Burk plots. We report the data as average ± s.e.m.

### Structural models for molecular dynamics (MD) simulations

A complete checklist and data for MD simulations are provided in Supplementary Table 6 and Supplementary Data 1. We have used two systems to perform SAXS-driven metainference metadynamics simulations of the isolated D1: the D1 group II intron in the "closed" and "catalytic" conformations as obtained from PDB id 4Y1O and 4FAQ, respectively. To use the same topology for both conformations and perform multiple-replica simulations, we modeled the "closed" state by performing a steered molecular dynamics (SMD) simulation starting from the "catalytic" structure. We used the RMSD relative to the "closed" state as a collective variable, which was linearly decreased to reach 0 nm value using a force constant k = -25 kcal•mol$^{-1}$. We used the same approach to generate the seed models for the simulations of the construct D1-2. In both cases, we replaced every Ca$^{2+}$ ion in 4FAQ with Mg$^{2+}$ to simulate catalytically compatible conditions. Importantly, the structural positions of these ions are strictly conserved across group II intron catalytic states[29,58], and we had already confirmed in previous studies that such ionic replacement does not affect the behavior of the molecule in MD simulations[23,30]. Nonetheless, here, to further check whether the replacement of Ca$^{2+}$ with Mg$^{2+}$ in 4FAQ accurately mimics the conditions that lead to capturing 4Y1O, we have performed additional equilibrium MD simulations starting from our modeled D1 in isolation (i.e., D1 coordinates from 4FAQ, Ca$^{2+}$ replaced with Mg$^{2+}$). Our simulations show that in these conditions, D1 spontaneously collapses into a closed conformation similar to that found in 4Y1O, suggesting that replacing Ca$^{2+}$ with Mg$^{2+}$ reproduces the behavior of D1 in isolation (Supplementary Fig. 4).

Similarly, we have used two systems for MD simulations: the D1-3 group II intron in the "partly open" conformation and in the "fully open" conformation obtained from cryo-EM experiments. Both molecular structures had unresolved residues. In the "partly open" structure, the missing residues were modeled outward from the helix. Since the "fully open" structure had a greater number of experimentally unresolved bases, we performed steered molecular dynamics (SMD) simulation starting from the "partly open" model. During the SMD, we used two collective variables (CVs). The first CV was the RMSD relative to the resolved residues of the "fully open" cryo-EM structure. The second CV utilized 20 configurations extrapolated from cryo-EM data using *CryoSPARC*. We employed these configurations to create a molecular path for the conformational transition under investigation.

We based this molecular path on the positions of the P atoms in the backbone of a portion of D1, using the PATHMSD CV implemented in PLUMED[59,60], defined as follows:

$$spath = \frac{\sum_{i=1}^{N} i e^{-\lambda R(X-X_i)}}{\sum_{i=1}^{N} e^{-\lambda R(X-X_i)}} \tag{1}$$

$$zpath = -\frac{1}{\lambda} ln\left[\sum_{i=1}^{N} e^{-\lambda R(X-X_i)}\right] \tag{2}$$

During the SMD, we decreased the RMSD linearly over the simulation time until it reached a value of 0 nm, using a force constant k = -25 kcal•mol$^{-1}$. In contrast, we kept the second CV, zpath, fixed at 0 using a force constant k = -500 kcal•mol$^{-1}$.

### MD simulations set up

For the parametrization of RNA, we used the AMBER-ff12SB (ff99 + bsc0 + χ$_{OL3}$), incorporating the Joung–Cheatham parameters[61] for monovalent metal ions (K$^+$) and the parameters by Li et al[62]. for divalent metal ions (Mg$^{2+}$)[63]. We enclosed each system within a box measuring approximately 127 x 129 x 113 Å$^3$, which was filled with a 20 Å layer of TIP3P water molecules. The simulation boxes contained approximately 59,000 water molecules, bringing the total atom count for each system to about 190,000. We added magnesium and potassium ions to neutralize the systems in a ratio of about 1:3 to mimic ionic concentrations of 50 mM for magnesium ions and 150 mM for potassium ions as used in our SAXS experiments. We first energetically minimized each molecular complex. To equilibrate each system, we sequentially performed 1 ns in NVT ensemble at 310 K, and 5 ns in NPT ensemble at 1 bar and 310 K simulations starting from each initial minimized structure. We controlled the temperature and pressure of all systems using the v-rescale thermostat[64] and Parrinello–Rahman barostat[65], respectively. We calculated the electrostatic interactions using the particle mesh Ewald method[66] with a real space cut-off of 11 Å. We set the cut-off value for van der Waals interactions at 11 Å. We performed the MD simulations with the GROMACS 2023 software package patched with PLUMED 2.9[60]. We performed analyses on production trajectories, i.e., after the equilibration of the systems.

### Metainference metadynamics simulations

We have performed metainference metadynamics simulations on the isolated D1 and the construct D1-2 with well-tempered, parallel-bias, and multiple walker protocols, using 6 parallel replicas starting from both the "catalytic" and "closed" conformations (three replicas, each)[67–70]. We have used 3 collective variables. The Z and S components of one path collective variable are defined by 15 equally spaced intermediate conformations of the nucleotides 71–73 and 115–116 (hinge 1), going from bulge-in to bulge-out states as captured by PDB id 4Y1O and 4FAQ, respectively. The third collective variable is represented by the R$_g$ of D1. We biased the S and Z components and the R$_g$ by depositing Gaussians of 0.3 kJ/mol height every 500 steps, using a bias factor of 15, and using a sigma of 0.5, 0.01, and 0.033 in the appropriate unit for each coordinate, respectively. Each replica spans -250 ns for D1, and -400 ns for D1-2, summing up to -3.9 µs total simulation time.

We calculated a set of 15 representative SAXS intensities at different scattering angles, ranging between 0.01 and 0.5 nm$^{-1}$ and equally spaced, with PLUMED using atomistic structure factors[71,72]. We applied SAXS restraints every 10 steps and used atomic scattering factors to back-calculate the 15 SAXS intensities. Both $\sigma_i^{SEM}$ and the $\sigma_{r,i}^B$ were automatically estimated. SAXS restraints were applied every 2 MD steps, and atomic scattering factors were used to back-calculate the 15 SAXS intensities[67].

## Multiple walker well-tempered metadynamics (MW-MetaD) simulations

To improve the sampling of the conformational space during the transition between the "partly open" and "fully open" state of the D1-3 system, we employed multiple walker well-tempered metadynamics (MW-MetaD) simulations. Specifically, we simulated two molecular systems for 1 $\mu$s of simulation time, one corresponding to the "partly open" state and the other to the "fully open" state. During MW-MetaD, the simulations shared the deposited bias, allowing for faster convergence. We applied the bias to the spath CV (see Eq. 1) based on a new path derived from the SMD simulations described above. The reference path involves only the positions of the P atoms in the backbone, which rearrange as defined by the SMD data, and uses the spath as the CV that tracks the progress of the system along the reference path. Constraining only the P atoms in the backbone provided freedom to the orientation of the nucleotide bases, allowing them to explore the conformational space freely. Additionally, we maintained the zpath CV (see Eq. 2), which accounts for the distance of sampled conformations from the reference path, below 0.2 nm (with a force constant k = -5000 kcal•mol$^{-1}$) to explore conformations closely aligned with the reference path. In this manner, the MW-MetaD simulations sampled the conformational space to identify the lowest energy pathway for the conformational change under examination. We constructed this new path by extracting 20 configurations evenly spaced along the RMSD variable relative to the wide-open conformation. The spath CV has a value of 1 in the "partly open" state and a value of 20 in the "fully open" state. We used a bias factor of 8 and a $\sigma$ = 0.5.

## Grid preparation for cryo-EM and cryo-EM data collection

We applied 1.6 µL of the samples on both sides of glow-discharged 300-mesh R 1.2/1.3 Quantifoil Cu (copper) grids. We blotted the grids for 1 s and then vitrified them in liquid ethane using a Vitrobot Mark IV (Thermo Fisher Scientific) with the chamber at a temperature of 22 °C and at 100% humidity. We acquired 26,214 micrographs of the D1-2 sample at ESRF CM01 at a dose rate of 38.07 e/Å$^2$. We acquired 16,719 micrographs of the D1-3 sample at the EMBL Heidelberg imaging center facility at a dose rate of 40 e/Å$^2$ on a 300 KeV Titan Krios with a Gatan K3 camera. Finally, we acquired 12,225 micrographs of the D1-4 sample at the EMBL Heidelberg Imaging Center facility at a dose rate of 40 e/Å$^2$ on a 300 KeV Titan Krios with a K2 camera.

## Cryo-EM data processing

We used *CryoSPARC* versions 3.0 and 4.0 for data processing. Processing consisted of motion correction, CTF estimation, manual curation, particle picking, extraction (250 Å box size), 2D classification, ab initio modeling, and 3D classification.

Upon obtaining a homogenous-refined single model, we used 3D variability analysis to obtain the ensemble of cryo-EM maps for the D1-2, D1-3, and D1-4 samples with 3 components and filter resolution settings at 4.0, 4.0, and 6.0 Å, respectively. We built models for the 20 3DVA cryo-EM maps of each construct, following the hierarchical rigid-body protocol outlined in ref. 73 using the X-ray structure PDB id 4FAQ as a starting model. We performed rigid-body restraints defined using RIBFIND2[74], and then flexibly fit the models into the cryo-EM maps using the software TEMPy-ReFF[75].

For highly-selective particle exclusion, we employed an in-house procedure to iteratively score, rank, and select targeted individual particles of the D1-2 and the D1-3 datasets. This approach aimed to maximize the proportion of high-quality particles using iterative scoring of each individual particle against re-projected map of each conformational state of group II intron domain constructs, followed by particle re-ranking and re-selection. We confirmed the overall enhanced quality of the selected particles by re-estimating positional and angular parameters using the non-uniform

refinement and 3D reconstruction in *CryoSPARC*. Then, we used coordinates extracted from PDB id 4FAQ as a starting model to manually fit into the sharpened cryo-EM density map obtained with DeepEMhancer. Successively, we performed real-space refinement in Phenix to obtain the map-to-model cross-correlation scores. We used these latter models to correct RNA geometry using the software ERRASER[76], and we finalized refinement with TEMPy-ReFF using the GMM (Gaussian mixture model) fitting mode (Supplementary Figs. 5–6).

For D1-4, due to the lower resolution of this dataset, after 3D classification, we have performed non-uniform refinement to obtain maps for the "partly-open" and the "fully-open" states. We then used the coordinates of D1-2 obtained as described above, and TEMPy-ReFF to flexibly fit the models into the cryo-EM maps. We then performed real-space refinement in Phenix to obtain the map-to-model cross-correlation scores and directly finalized refinement with TEMPy-ReFF using the GMM fitting mode (Supplementary Fig. 7).

We validated each of the resultant atomic models using TEMPy SMOC score[73] to assess nucleic acid fit-to-map quality (averages across nucleic acids are reported in Table 1) and using Phenix[77] to assess overall fit-to-map and validate geometry.

All distances and angles reported in the text were measured between independently refined atomic models, not directly from raw density. Helical axes were defined by a least-squares fit of all sugar–phosphate backbone atoms to a central line[78,79].

## Inclusion and Ethics Statement

This research aligns with the Inclusion & ethical guidelines embraced by Nature Communications.

## Reporting summary

Further information on research design is available in the Nature Portfolio Reporting Summary linked to this article.

# Data availability

Coordinates and EM micrographs have been deposited in the Protein Data Bank and Electron Microscopy Data Bank under accession codes: 9G4I and EMD-51040 ("partly open" state of D1-2), 9G4V and EMD-51068 ("fully open" state of D1-2), 9G4L and EMD-51044 ("partly open" state of D1-3), 9G4J and EMD-51041 ("fully open" state of D1-3), 9G54 and EMD-51077 ("partly open" state of D1-4), 9G56 and EMD-51080 ("fully open" state of D1-4). SAXS data have been deposited in the Small Angle Scattering Biological Data Bank under accession codes: SASDX39 (D1), SASDX59 (D1-2), SASDX49 (D1-3), and SASDX69 (D1-4). Molecular dynamics simulation files have been deposited in Zenodo, entry 17100464 and are provided as Supplementary Data 1. Data supporting the findings of this study are available within the manuscript and its Supplementary Information files. Source data are provided with this paper. Newly created plasmid DNAs are available from the corresponding authors upon request. Source data are provided with this paper.

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

## Acknowledgements

We thank the scientists of the EMBL Grenoble, EMBL Heidelberg and ESRF CM01 cryo-EM facilities for assistance and support in cryo-EM data collection. We thank the scientists of the EMBL Grenoble and ESRF for assistance and support in SAXS data collection at beamline BM29. We thank Ombeline Pessey and Dr. Ines Dieryck for excellent technical assistance. We thank Dr. Catarina Silva, Dr. Benoit Arragain and Dr. Luciano Dolce for helpful discussion and advice on cryo-EM data processing. We thank all members of the Marcia, Topf and De Vivo labs for helpful discussion. Work in the Marcia lab has been partly funded by ITMO Cancer (18CN047-00), Région Auvergne Rhône Alpes (project R21105CC; allocation RPH21004CCA), FINOVI (AAP15), Canceropole CLARA (Oncostarter) and by the Fondation ARC pour la recherche sur le cancer (PJA-20191209284) and is now partly funded by the Swedish National Research Council (VR, 2024-04107), by the HORIZON-MSCA-2023-DN-01 action (project: TargetRNA, n. 101168667), and by the Italian Association for Cancer Research (AIRC, IG 28746). The Marcia lab uses the platforms of the Grenoble Instruct-ERIC center (ISBG; UAR 3518 CNRS-CEA-UGA-EMBL) within the Grenoble Partnership for Structural Biology (PSB), supported by FRISBI (ANR-10-INBS-0005-02) and GRAL, financed within the University Grenoble Alpes graduate school (Ecoles Universitaires de Recherche) CBH-EUR-GS (ANR-17-EURE-0003). The Topf lab thanks the Leibniz Institute of Virology as part of Leibniz ScienceCampus InterACt (funded by the BWFGB Hamburg and the Leibniz Association), a Wellcome Collaborative Award in Science (209250/Z/17/Z), the 703 Landesforschungsförderung Hamburg (HamburgX) and Deutsche Forschungsgemeinschaft (DFG) through CRC 1648 (Project B01). Marco De Vivo thanks funds through the project "National Center for Gene Therapy and Drugbased on RNA Technology" (CN00000041), financed by NextGenerationEU PNRR MUR – M4C2 – Action 1.4 – Call "Potenziamento strutture di ricerca e di campioni nazionali di R&S" (CUP: J33C22001130001). Marco De Vivo also thanks the Italian Association for Cancer Research (AIRC) for financial support (IG 30631).

## Author contributions

Marco Marcia conceived, designed and supervised the work and obtained funding; Shekhar Jadhav cloned, produced, purified and biochemically and enzymatically characterized all *O. iheyensis* group II intron constructs, obtained all SAXS datasets, and acquired and processed all cryo-EM datasets, including refinement of the corresponding structures; Jacopo Manigrasso, Stefano Muscat, and Marco De Vivo performed and analyzed the MD simulations; Mauro Maiorca, Thomas Mulvaney and Maya Topf helped with particle ranking and selection protocol to obtain the maps of D1-2 and D1-3, as well as model refinement and assessment; Spandan Saha and Auriane Rakitch cloned, produced, purified and enzymatically characterized ai5y-D135 and ai5y-D135-hinge1mut; Shekhar Jadhav and Marco Marcia interpreted the data

with contribution of all authors; Shekhar Jadhav and Marco Marcia produced the first draft of the manuscript; all authors approved the final version of the manuscript.

## Funding

## Competing interests

The authors declare no competing interests.
