## [Transparent Peer Review file · Nature Communications]

Dynamic assembly of a large multidomain ribozyme visualized by cryo-electron microscopy

Corresponding Author: Dr Marco Marcia

Version 0:

Reviewer comments:

Reviewer #1

(Remarks to the Author)

In this study, Jadhav, et al. used cryo-EM, SAXS and MD simulations to study the dynamics of the group II intron D1 domain alone and in the presence of 3' domains (D2-D4). The authors categorized D1 dynamics into the "closed", "partly open" and "open" states, quantified by angles between helices and "gate" distances that ultimately affected D5 docking. In general, the article was not well organized. The authors presented a lot of tedious and unnecessary information in the text and figures, diminishing the major findings in this study. The reviewer is not convinced that the SAXS data added any value to this work, MD simulations could also be performed based on the cryo-EM density. While it was not clear if the angles were useful to quantify the D1 dynamics, the gate distance measurement was actually not valid. The authors likely didn't quite understand the definition of resolution, so they kept trying to interpret distance changes much smaller than the achieved cryo-EM resolution, which was completely invalid. For instance, for the 4 Å cryo-EM maps, changes smaller than 4 Å should not be described, as such changes could not be captured under such resolution.

Another major concern is that the authors derived the sequential folding of D1-4 by determining cryo-EM structures of D1-2, D1-3 and D1-4. While these truncated constructs can never represent the actual folding status of the full-length intron, the sequential folding can never be validated as no timing was involved in either SAXS or cryo-EM. A more appropriate experiment design that may overcome this flawed logic is to determine the full-length group II intron and to classify the possible folding dynamics. Obtaining high-resolution group II intron structure has been recently proved possible (Haack, et al., 2025). In conclusion, the current study appeared to be very preliminary and was not ready for publication. Other detailed criticisms are as follow:

1. The authors should avoid crystallography terms such as reciprocal orientation when describing cryo-EM structures, as they are invalid in cryo-EM analysis.
2. The authors presented cryo-EM structures in Fig. 3, 4 and 6 as if they have not built models, however, data availability section indicated that there were models built for each map. In this case the authors should then described cryo-EM findings with maps and models together.
3. References are generally not properly selected. Line 4, ref 1 and 2 are reviews written by the authors that focus on RNA-targeted small molecules. While this is an important and continually growing field, it is still way too preliminary to represent "RNAs with important implications for human health"; Line 10, another misfolded Tetrahymena ribozyme by Li, et al., 2022 was omitted;
4. To the best of my knowledge, folding kinetics of group I introns have been more systematically explored comparing to group II introns. I have no objection that understanding group II intron folding kinetics is important, but the authors should include an extra paragraph about group I intron (and other RNAs, e.g. PMID 30275276 and refs therein) folding in the introduction to brief the readers with the current status of RNA folding.
5. In general, I find this article hard to read through, e.g. Page 3, line 29-35; Page 4, line 7-9; Page 8, line 17-21, what does reciprocal orientation stands for in cryo-EM structures?
6. Aren't α , β , etc. Greek letters? I suggest using other systems for angle annotation because Greek letters have been used

as tertiary interaction nomenclatures in group II introns.

7. I strongly suggest including a cartoon panel in Fig. 1 to give the readers an overall picture of the D1 domain and where should they pay attention to. It is very hard to follow all the angle details in the current ribbon diagram.

8. I think all figure panel should be described in the main text. For example, the authors described Fig. 1G-H directly after Fig. 1A, are panels B-F not important? If so why are they in the main figure anyway? This problem exists in almost all figures.

9. In my opinion, the authors presented this study in a rather tedious way. For example in Fig.1, the only important message seemed to be just comparison of D1 domains in the closed and catalytic states showed i1-i2 helix clashed with D5. What's the point of measuring all those angles?

10. If Supplementary figure 16 appears so early, maybe it should be Supplementary figure 5?

11. While it is glad to see that authors are aware of cryo-EM density as Coulomb potential rather than electron density, this does not need to be repeated through out the manuscript. "Density" is simple and accurate enough.

12. Page 2, line 6: Change molecular simulations to molecular dynamics simulations; Page 34, Fig 6D: Change D1-3 "fully open" to D1-3 "partly open"; Page 42-44: There are some terms not unified in the 3 Cryo-EM schematics. Also, the figures failed to include how the maps were generated after the 3DVA step; Page 46, SFig 9B: duplicated residue 121. There are multiple "D12", which is probably "D1-2"?

Reviewer #3

(Remarks to the Author)

Jadhav et al describe in their manuscript a detailed study on the domain assembly of a group II intron ribozyme, a most complex and large RNA machinery. Self-splicing group II introns are closely related and the evolutionary ancestors of the eukaryotic splicing machinery. From this perspective the topic is highly important not only for ribozyme specialists but for a much wider community and hence merits publication in Nature Communication. Nevertheless, we find several major issues with this manuscript, which should be addressed in detail, before this manuscript could be considered for publication in Nature Commun:

Generally, the manuscript is tailored very strongly towards experts in group II intron biochemistry. Those readers not so familiar with group II introns will have problems following the argumentation and judging the results in the correct way. For example: the study does not provide information on the catalytic activity of the constructs used (D1-2, D1-3, and D1-4), while only the D1-5 construct is known to be catalytically active. Including this information, either through experimental validation or literature evidence, would strengthen the interpretation of the observed folding pathway.

This is also reflected by the authors going very much into detail in the text with distances between specific nucleotides/regions, or angles between helices, etc, which makes it rather difficult and tedious to follow the overall story. The amount of detail is appreciated, but the authors might find a better way, to provide the reader with the numbers without including them in every sentence. Maybe they could also concentrate on the most important findings, adding further details in the Supplementary Material.

By SEC-SAXS that authors have experimentally and convincingly verified the overall size of the structures obtained by MD simulations. However, they also go into much detail discussing conformations of single nucleotides, solely based on simulations, and without experimental evidence (e.g. mutational analysis or others), for example on page 6/middle paragraph, but also elsewhere.

p 12. The energies obtained by FES for the "partly open" and "fully open" states (2 and 4 kcalmol⁻¹). How meaningful are these energy values? What do they mean in terms of an equilibrium/distribution between these two states? The authors only talk about these two minima in the energy landscape but not on the transition in-between them, i.e. the activation barrier. Hence, the statement "This rather flat FES indicates that conformational transitions are energetically inexpensive ..." is actually such not correct, because the transition is not dependent on the values of the minima, but on the height of the activation barrier in between. In fact, as the authors actually find both states in the cryoEM, this indicates, that the barrier is actually rather high. Please modify accordingly.

Discussion: The discussion is very weak, as it fails to connect the findings to the larger RNA world. It lacks depth and needs to be placed in a broader context—what do the results imply for the overall folding pathway? Are the newly discovered conformations essential intermediates? How might these findings relate to other introns or RNAs, like the spliceosome? Additionally, highlighting open questions and providing a more global perspective will enhance the discussion and demonstrate its relevance to a wider scientific community.

Further points in the Discussion:

First sentence (p13, l2-4): The statement that the data visualizes for the first time the multi-domain assembly pathway while "avoiding unproductive kinetic traps" should be further discussed and/or revised. While cryo-EM is indeed a powerful technique for studying RNA folding and revealing kinetic traps (Bonilla, Science Advances, 2022), the absence of observed misfolded structures does not necessarily indicate the absence of kinetic traps. Several factors may contribute to the lack of detected misfolded states, including the transient nature of kinetic traps, conformational heterogeneity and dynamics, effects of vitrification during sample preparation, resolution limitations, and particle selection bias. Given these considerations, these limitations need to be discussed in more detail to reflect the possibility that undetected misfolded states may still be present.

p14, l3-4). This sentence does not fit well in its current position, as the following sentence does not discuss other RNAs or how the insights gained can be applied more broadly. It would be more appropriate in the next section (same page, line 28), where it aligns better with the context.

p14, l5 – 27: It is unclear what the two newly visualized states, "partly open" and "fully open," represent in the context of RNA folding. A more detailed discussion and interpretation is needed. Are both states essential intermediates in the folding pathway? Under what conditions would one state be favored over the other? Do the two intermediates serve distinct functional roles?

p14, l9 - 22: The description of the D1 transition is unclear.

a) The text states that multi-domain assembly is not accomplished through sequential conformational changes of D1, INSTEAD, D1 transitions from a closed state to partly open and fully open. The latter suggests a sequential pathway, so there is no contradiction - please elaborate on this point?

b) However, figure 7 indicates that D1 can adopt either the partially open or fully open state, raising the question of which description is now correct. Does the transition take place sequentially or one after the other?

c) Furthermore, if D1 can adopt either the partially or fully open state, what determines which conformation is formed? And if the two open states are interchangeable, what is the trigger for this transition?

p14, l25 – 27: Also here, please elaborate more on this dynamic gating system with regard to a large D4. Describe how D1 should look like so that D4 can be properly docked and the ORF is not interfering

The Video is well made and effectively helps to visualize the process. However, in some places, additional labelling would improve clarity, and some transitions happen too quickly, making it difficult to follow. Below are specific suggestions for improvement:

- Beginning: Indicate which conformation is being shown.
- Second 6: Clarify what the yellow conformation represents.
- Second 16: This transition is too fast and should be slowed down.
- Second 34: Label the blue domain for clarity.
- Second 38: Specify which conformation the blue Coulomb potential density corresponds to, and indicate which constructs are being shown.
- Second 70: Label the purple domain and the later-appearing purple density.
- Second 70: Clearly indicate which constructs the purple structures correspond to (D1-3).
- Second 104: Label the orange construct (D1-4) for consistency.
- Second 122: In the final image, name the domains and indicate the conformation, as this represents the entire structure. Adding these details would enhance the video's clarity and ensure that viewers can fully understand the structural changes being depicted.

Comments to Figures:

Figure 1a: (i) Since this manuscript is intended for a broader audience beyond the group II intron community, the structural deviations from the wild-type construct must be mentioned/shown, i.e. the absence of D6 and the truncation of D2, D3, and D4. In the end, the authors are using a heavily modified construct and this will help readers unfamiliar with these modifications better understand the construct.

(ii) Please indicate the two exon binding sites and clarify whether the intron binding site is located on the 5' exon to provide additional structural context.

Supplementary Figure 4 – SEC Profile: (i) C-F: Including the calculated molecular weight (MW) of each construct would help clarify the interpretation of the mass photometry data.

(ii) C-F: The presence of two peaks in the SEC profiles is not explained—please provide an explanation for this observation.

(iii) In panels 4C and 4D, the main peak appears to have the same MW despite representing two different constructs. Could this be an error, or is there a specific reason for this result? Clarification would be appreciated.

Supplementary Figure 9: (i) There is an inconsistency between the text (p.15, l.17), which refers to "hinge 1," and the figure, which labels "hinge 2." Please clarify which is correct and ensure consistency.

Reviewer #4

(Remarks to the Author)

The authors combine single-particle cryo-em, SAXS, and molecular dynamics simulations to illustrate the multi-domain

assembly of a self-splicing group II intron – a system of considerable biomedical significance. The work is highly innovative and addresses a fundamental problem of multi-domain assembly of RNA into a functional state. Specifically, the work sheds light on important intermediates in the assembly pathway to achieve the catalytic state, and generally demonstrates how RNA domains may strategically assemble to avoid kinetic traps. The provided movie is especially helpful in understanding this complex process. This work constitutes a large-scale interdisciplinary collaboration that is, to my knowledge, completely novel in the insights that have been gained. The work is likely to be of broad interest to the community and readership of Nature journals.

There are a few clarifications and concerns that should be addressed before publication can be recommended:

1. A clearer description of how different structural data was used throughout this study is needed. In the paper, the authors compare geometric properties of crystal structures of the "closed" state (PDB ID 4Y1O) and "catalytic" state (PDB ID 4FAQ) from the referenced 2012 Cell paper. 4FAQ was crystallized with Ca²⁺ and it forms a dimer in the crystal. 4Y1O was crystallized with Mg²⁺ and is a monomer in the crystal. The following is my understanding of what the authors did (but it was not completely clear due to lack of details). Simulations depart from 4FAQ (catalytic) and do steered MD to get the "closed" state structure. The simulations are done with Mg²⁺, but supposedly depart from a structure containing Ca²⁺ with no description of these being replaced by Mg²⁺ or how that was achieved. Later in the paper the authors mention the catalytic state as 4FAR instead of 4FAQ, but it could be a typo because that is the only instance I saw it mentioned. That said, 4FAR is a structure from the same 2012 paper, but it is crystallized with Mg²⁺ and is a monomer. This does seem like a more appropriate structure to make comparisons with, so it left me questioning whether which structure they used in which instance. I think this illustrates that the manuscript would benefit with a clear summary of what structure(s) were used in what instances, and to provide additional details of the simulation set up. Some of the comments below are based on my assumption of what the authors did.

2. Page 5: The authors compare angles and displacements between structures of the "closed" and "catalytic" states from PDB's 4Y1O and 4FAQ. The 4Y1O (closed) structure was crystallized as a dimer, while the 4FAQ (catalytic) structure was crystallized as a monomer. Are the authors concerned that crystal packing interactions present in the dimer may impact these observations?

3. Page 9 line 29: The catalytic conformation is listed as PDB ID 4FAR. All other instances are 4FAQ. Is this correct? Both are structures of *Oceanobacillus iheyensis* group II intron but with different conditions. Does 4FAR make more sense? Computational details:

4. Page 18, Structure models for MD simulations: When preparing the systems (D1 etc) for SAXS-driven metadynamics metainference simulations, the authors started from the catalytic state and obtained the closed state from steered molecular dynamics. It is unclear how the placement of crystallographic metal ions, which are crucial for RNA folding and function [JPCB 126 (32), 5982-5990], were handled. The catalytic structure contains Ca²⁺ and K⁺ ions while the closed structure contains Mg²⁺ and K⁺ ions. Were the Ca²⁺ ion positions replaced with Mg²⁺, or were Mg²⁺ ions randomly placed to achieve the desired concentration? To what degree are the positions of these ions conserved between the states? The placement of Mg²⁺ ions in the simulations are very important, as they are important to assembly, folding and function. Further, the model for Mg²⁺ ions used by the authors, if their description is correct, is quite poor with respect to binding properties with RNA (see below). One could make predictions about Mg²⁺ ion binding sites for a given initial structure using 3D-RISM such as in [JACS 141 (6), 2435-2445]. Could such a procedure be validated by the structure with Mg²⁺ and then applied to other states?

5. Page 19, MD simulation setup: The authors state that the TIP3P water model was used and Mg²⁺ ions were treated with parameters by Li et al. The divalent ion parameters of Li et al. were developed in bulk water and not balanced with their interactions with nucleic acids. This has been demonstrated to be important for the more advanced 12-6-4 models used with TIP4Pew water and these are what is recommended for RNA simulations, see [JPCB 119 (50), 15460-15470] (the authors might want to consider this in the future – it is important if one is making predictions about divalent metal ion binding sites). Additionally, it is not clear whether the total ion concentrations of Mg²⁺ and K⁺ (100 mM and 150mM) were balanced with counterions to produce a neutral system and what parameters were used (perhaps I missed this detail).

6. Page 12 paragraph 2: Details of system preparation for the "partly open" and "fully open" metadynamics simulations should be provided in the supporting information.

7. SI figure 1A: The y axis label is missing a unit. Should the x axis label be residue?

Reviewer #5

(Remarks to the Author)

Version 1:

Reviewer comments:

Reviewer #1

(Remarks to the Author)

The revised manuscript by Jadhav, et al. is substantially improved. Including additional biochemical validation of hinge 1 mutagenesis is informative. The revised content is still somewhat hard to follow. The figure panels are not included in all figure citations in the text, making the entire manuscript hard to focus and follow.

Initially, my major problem is understanding the logical connection between structures of gradually longer constructs (D1, D1-2, D1-3, D1-4) and the sequential order of this assembly, as described in P10 Line 7 – 34. Later in the Discussion in P14 Line 28-36, this seems to be a reasonable explanation, which I strongly suggest to move to the corresponding paragraph in the Results section. The Discussion can simply keep the “first come, first fold”.

SAXS measures structural features in solution, which is indeed complementary to cryo-EM. However, I am not convinced that SAXS-driven simulations initiate or validate the working hypothesis of this study. SAXS results here represent the average globular shapes of different RNA constructs in solution. I cannot imagine the SAXS results indicate the presence of “closed”, “partly open” and “fully open” conformations that facilitate subsequent MD simulations and high-resolution cryo-EM analysis. Neither do I acknowledge that SAXS results validate the existence of these conformations in solution. Maybe the X2 values mentioned by the authors can be correlated to distribution of different conformations, but unless explicitly explained in the main text, I suggest to leave SAXS in the SI simply as in-solution results.

Energy barriers between conformations are interesting topics, the authors may consider including relevant literatures (e.g. <https://doi.org/10.1038/s41467-022-29332-2>) discussing the energy landscape preserved by cryo-EM. These energy numbers, although estimated, could explain why such conformations are resolved.

Other minor comments:

1. P3 Line 6 should probably name what are the two main pathways?
2. labelling the critical hinge 1 and 2 in Fig 1C-G helps the readers to visualize them in 3D.
3. P5 Line 13 – 19: When claiming the “least” structurally-displaced residues, is it actually that both structures were aligned to D1d1? This should be specified.
4. P5 Line 24: I am guessing the clash is referring to Fig. 1I, can this clash be highlighted so that readers know what to look at.
5. P6 Line 7: Make the right panel of Fig. 1G a separate panel, as only this panel contains D5, and highlight the clash between D5 and closed i1-i2, so that readers know what to look at. Cite the movie as well.
6. Fig 6A-D: These panels are trying to highlight the density supporting C116 and A72, but it's really not clear what should be looked at. Are the dashed circles suggesting missing density, which corresponds to bulged out conformation?

Reviewer #3

(Remarks to the Author)

Jadhav et al have extensively revised their manuscript according to the reviewers comments. My previous comments and issues have now all been attended to and sufficiently clarified. As stated previously, the topic is highly important not only for ribozyme specialists but for a much wider community, and hence I support now publication in Nature Communication.

Reviewer #4

(Remarks to the Author)

The authors have adequately addressed to concerns raised in the previous review and revised the manuscript accordingly.

RESPONSE TO REVIEWERS' COMMENTS FOR:

Dynamic assembly of a large multidomain ribozyme visualized by cryo-electron microscopy

Shekhar Jadhav^{1,7}, Mauro Maiorca^{2,3,4,#}, Jacopo Manigrasso^{5,6,#}, Spandan Saha⁷, Auriane Rakitch⁷, Stefano Muscat⁵, Thomas Mulvaney^{2,3,4}, Marco De Vivo^{5,*}, Maya Topf^{2,3,4,*}, Marco Marcia^{1,7,8,9,*}

*To whom correspondence should be addressed.

E-mail: marco.devivo@iit.it; maya.topf@cssb-hamburg.de; marco.marcia@icm.uu.se

We would like to thank all the reviewers for their careful assessment of our manuscript and for their comments and criticism. In this detailed point-by-point response, we have addressed all concerns of the reviewers. We also enclose a revised version of the manuscript and figures, in which revisions are marked in red.

Responses to reviewer #1:

In this study, Jadhav, et al. used cryo-EM, SAXS and MD simulations to study the dynamics of the group II intron D1 domain alone and in the presence of 3' domains (D2-D4). The authors categorized D1 dynamics into the "closed", "partly open" and "open" states, quantified by angles between helices and "gate" distances that ultimately affected D5 docking. In general, the article was not well organized. The authors presented a lot of tedious and unnecessary information in the text and figures, diminishing the major findings in this study.

Response 1-1

In response to this comment and to comments of reviewers 3-5, we have extensively reorganized and simplified our manuscript, for instance in the Introduction (page 3, lines 8-28; and page 4, lines 8-12); in Results (page 5, lines 33-37; page 6, lines 1-7 and lines 9-33; page 7, lines 15-32; page 8, lines 23-25; page 9, lines 8-15; page 10, lines 23-34; and page 11, lines 10-23); and in Discussion (page 14, lines 2-9, lines 15-18, and lines 21-37; page 15, lines 1-3, lines 6-14, and lines 17-34; and page 16, lines 10-21).

The reviewer is not convinced that the SAXS data added any value to this work, MD simulations could also be performed based on the cryo-EM density.

Response 1-2

In our manuscript, we have integrated MD simulations with both SAXS and cryo-EM data. SAXS-driven MD simulations are described in the main text on page 6, lines 19-33 and page 7, lines 17-32 (**Figure 2, Supplementary Figure 4 and Supplementary Table 2**). Cryo-EM-driven MD simulations are described in the main text on page 11, lines 24-37, and page 12, lines 1-17 (**Figure 7**).

We do agree with the reviewer that cryoEM-driven simulations are the most important for our study, because they support our mechanistic model for group II intron multi-domain assembly. But in line with the assessments of reviewers 3-5, we would also like to politely ask to maintain the SAXS-driven simulations, which are complementary to the cryoEM-driven one and which are essential for the completeness of our work because of the following considerations. First, SAXS-driven MD simulations are an important checkpoint to model our system as it behaves in solution, rather than vitrified on EM grids (as described on page 7, line 20-21, and on page 8, lines 8-10). Second, coupling SAXS data with MD simulations for D1 helped us define a working hypothesis that we then explored by cryo-EM (as described on page 6, lines 34-37; on page 7, lines 1-4; and on page 7, lines 33-34). Third, SAXS-driven MD simulations coupled to 3DVA analysis by cryo-EM of D1-2 and D1-3 constructs help us improve the description of our system in solution, as it emerges from the improved X² values (page 9, lines 16-19, and

Supplementary Figure 4 and Supplementary Table 2).

While it was not clear if the angles were useful to quantify the D1 dynamics, the gate distance measurement was actually not valid. The authors likely didn't quite understand the definition of resolution, so they kept trying to interpret distance changes much smaller than the achieved cryo-EM resolution, which was completely invalid. For instance, for the 4 Å cryo-EM maps, changes smaller than 4 Å should not be described, as such changes could not be captured under such resolution.

Response 1-3

We thank the reviewer for raising this point and apologize for not making sufficiently clear in our initial draft that the conformational changes that our study captures are very pronounced and go well beyond the resolution limits of our cryo-EM datasets. The conformational changes of helices D1c and D1d1 displace nucleotides in the terminal loops of these helices (i.e. residue C91) by 51.2 Å between the “fully open” state and the “partly open” state! We have now added this information in the main text (page 9, lines 34-35), and in **Figure 6**.

Here, we would like to clarify the rationale for the choice of the specific descriptors that we have selected in our study. Importantly, we want to stress that the use of angles and distance parameters as a mean to provide a structural description of conformational changes in biological macromolecules is a routinary approach, previously used in the literature for many other targets, including both proteins (Thangappan et al., 2017) and nucleic acids (Bailor et al., 2011; Lietzke et al., 1996). To justify our rationale, we have now added this information and references in the main text (page 5, lines 34).

1) Angles

To describe the geometry of RNA structures, the use of angles between RNA helical axes is well established (Bailor et al., 2011).

This type of descriptor is very useful in our study, where the most pronounced conformational changes that we observe are near-rigid body movements of helical motifs. Specifically, the cryo-EM structures of D1-2, D1-3, and D1-4 show a distinct change in the angular orientation of the D1c helix with respect to the other helices that originate from the same 5-way junction characteristic of the D1 scaffold. That is why, to characterize this distinct angular orientation, we have introduced the measurement of ‘*angle A*’ between the helical axis of D1c and D1d1 (**Figures 1 and 4**, and **Supplementary Table 1**). ‘*Angle A*’ captures exactly how the D1 core switches from its “closed”, to its “open” and finally to its “catalytic” states. Therefore, it provides the description of the trajectory that D1 needs to follow to transit from non-functional states to a functional conformation. The other angles that we have used serve analogous purposes to describe the reciprocal orientations of the two sections of helix D1c (core vs peripheral, ‘*angle B*’), of D1c vs D1_{i1-i2} (‘*angle C*’), and of D1d1 vs D1_{i1-i2} (‘*angle D*’, **Figure 1** and **Supplementary Table 1**).

2) Gate distance

While the angles described above are the most robust descriptors of our observed RNA conformational changes, we have decided to additionally use an interatomic pairwise distance descriptor. This distance provides a tangible direct measurement of the size of the D1 cavity, which is what determines the docking of the active site residues contributed by D5 in the catalytic state.

Depending on how far from the pivot point (i.e. how far from the junction between D1c and D1d1) such distance is measured, its dimension changes, for obvious geometrical reasons, and we believe that this is what may have created confusion for the reviewer. Close to the pivot point (i.e. between residues G75 from ‘core D1c’ helix and U238 from D1d1 helix), the gate distance is of about 36 Å and it changes by ± 4 Å between the “closed”, “partly open”, “fully open” and “catalytic” states (**Figure 4**). However, at the tip of these helices (i.e. between residue C91 in D1c and residue U150 in D1d1), the gate distance is of about 104.6 Å and 75.4

Å in the “fully open” and “partly open” state, respectively. This distance changes by 51.2 Å between the “partly open” and the “fully open” states (see main text on page 9, lines 34-35; and **Figure 6**). Therefore, it is evident that the distance changes that we observe in our structures are significantly bigger than the resolution limits of our structures.

Finally, we are aware and agree with the reviewer that in cryo-EM each map has an inherent spatial resolution limit, defined as the smallest separation between density features that can be reliably distinguished from noise, operationally identified as the inverse of the highest spatial frequency up to which the Fourier Shell Correlation function between two independently reconstructed half-dataset remains above a specific correlation value. We also agree that within a single map, positional measurements or distance evaluations smaller than the nominal resolution threshold become unreliable, as such features cannot be confidently differentiated from the intrinsic background variations in the map (Chen et al., 2013). However, we would like to politely point out that when comparing two cryo-EM maps of the same biological structure captured in distinct conformational states, distance measurements are typically made between defined extended structural reference landmarks present across all investigated conformations rather than by solely evaluating isolated density regions. These landmarks – such as the axis representing the backbone of an alpha-helix in proteins, or the sugar-phosphate backbone in nucleic acids – are determined through a geometric optimization procedure using multiple atomic coordinates, commonly the alpha-carbon (C_{α}) positions in proteins or atoms of the ribose moiety of nucleobases in nucleic acids (Enkhbayar et al., 2008; He et al., 2015; Lu & Olson, 2003; Mohan et al., 2014; Mohan & Noller, 2017). For instance, the axis for an alpha-helix can be derived by minimizing the sum of squared perpendicular distances of each C_{α} atom from a central fitted line (Pettersen et al., 2021). Because this axis represents a statistically fitted reference, its positional precision generally surpasses the nominal resolution limit of the individual cryo-EM maps. Consequently, comparative analyses between optimized reference points across two maps can accurately detect positional shifts smaller than the nominal resolution of each map, provided that methodological consistency is maintained. For these reasons, in **Figure 4**, we prefer to indicate the distance close to the pivot point (i.e. between residues G75 from ‘core D1c’ helix and U238 from D1d1 helix), which we can measure for all 3DVA states of our target (the tip is flexible and thus not visible in the map in some of the 3DVA states).

Another major concern is that the authors derived the sequential folding of D1-4 by determining cryo-EM structures of D1-2, D1-3 and D1-4. While these truncated constructs can never represent the actual folding status of the full-length intron, the sequential folding can never be validated as no timing was involved in either SAXS or cryo-EM. A more appropriate experiment design that may overcome this flawed logic is to determine the full-length group II intron and to classify the possible folding dynamics. Obtaining high-resolution group II intron structure has been recently proved possible (Haack, et al., 2025). In conclusion, the current study appeared to be very preliminary and was not ready for publication.

Response 1-4

We thank the reviewer for raising this concern.

First, we would like to clarify that we have solved and deposited 43 full-length group II intron structures until now, including 33 crystal structures (Manigrasso et al., 2020; Marcia & Pyle, 2012; Silvestri et al., 2024) and 10 cryo-EM structures (Kretsch et al., 2025; Topf et al., 2025). However, these datasets only capture the “catalytic” state of the target, not their assembly intermediates. The creation of our truncated D1-2, D1-3, and D1-4 constructs was therefore the only way to dissect the molecular mechanism by which the group II intron domains assemble sequentially onto their domain D1 scaffold. Our data show that the domains assemble in a sequential – and functionally relevant – order, ensuring that peripheral domains like D4 do not lead to the formation of kinetic traps that would otherwise compromise the proper assembly of the intron catalytic pocket. By resolving the structures of these constructs,

our work also unprecedentedly reveals at the molecular level the contribution of each structural 3'-terminal domain onto the conformation of the D1 folding scaffold of group II introns, enabling the identification of novel conformations, which would have otherwise been impossible to observe.

Second, we would like to remark that our structural observations have been possible thanks to our use of a dedicated cryo-EM processing pipeline, that enables capturing RNA dynamics at molecular resolution. Thus, besides the functional insights into group II intron assembly, our work also provides an important methodological advance in RNA cryo-EM.

Last but not least, our work uniquely integrates *protein-free* RNA cryo-EM with *in vitro*, *in solution* and *in silico* techniques such as biochemical and biophysical assays, SAXS, and multi-microsecond MD simulations, and thus provides a comprehensive multi-faceted description of a large RNA system, such as the group II intron ribozymes.

Other detailed criticisms are as follow:

1. The authors should avoid crystallography terms such as reciprocal orientation when describing cryo-EM structures, as they are invalid in cryo-EM analysis.

Response 1-5

We have replaced the term “reciprocal” with “relative” on page 5, line 29; and on page 8, lines 22 and 34.

2. The authors presented cryo-EM structures in Fig. 3, 4 and 6 as if they have not built models, however, data availability section indicated that there were models built for each map. In this case the authors should then described cryo-EM findings with maps and models together.

Response 1-6

We have modified **Figures 3, 5 and 6** according to this suggestion from the reviewer.

3. References are generally not properly selected. Line 4, ref 1 and 2 are reviews written by the authors that focus on RNA-targeted small molecules. While this is an important and continually growing field, it is still way too preliminary to represent “RNAs with important implications for human health”; Line 10, another misfolded Tetrahymena ribozyme by Li, et al., 2022 was omitted;

Response 1-7

We have followed the recommendation of the reviewer and have included the indicated reference on page 3, line 20 (Li et al., 2022). We have also added new references on page 3, line 4 to support the consideration that RNA controls vital cellular pathways, and that these crucial functions of RNA (rather than RNA drug targeting) are important for human health (Cech, 2002; Morris & Mattick, 2014).

4. To the best of my knowledge, folding kinetics of group I introns have been more systematically explored comparing to group II introns. I have no objection that understanding group II intron folding kinetics is important, but the authors should include an extra paragraph about group I intron (and other RNAs, e.g. PMID 30275276 and refs therein) folding in the introduction to brief the readers with the current status of RNA folding.

Response 1-8

To address this comment of the reviewer, we have included the indicated reference (Herschlag et al., 2018) and a completely new paragraph with examples of folding studies on RNAs of different classes in the Introduction (page 3, lines 8-20) and in the Discussion (page 14, lines 2-9).

5. In general, I find this article hard to read through, e.g. Page 3, line 29-35; Page 4, line 7-9; Page 8, line 17-21, what does reciprocal orientation stands for in cryo-EM structures?

Response 1-9

To address this and previous comments of this reviewer (see above), we have extensively reorganized and simplified our manuscript, for instance in the Introduction (page 3, lines 8-28; and page 4, lines 8-12); in Results (page 5, lines 33-37; page 6, lines 1-7 and lines 9-33; page 7, lines 15-32; page 8, lines 23-25; page 9, lines 8-15; page 10, lines 23-34; and page 11, lines 10-23); and in Discussion (page 14, lines 2-9, lines 15-18, and lines 21-37; page 15, lines 1-3, lines 6-14, and lines 17-34; and page 16, lines 10-21).

We have also replaced the word “reciprocal” with “relative” on page 5, line 29; and on page 8, lines 22 and 34.

6. Aren't α , β , etc. Greek letters? I suggest using other systems for angle annotation because Greek letters have been used as tertiary interaction nomenclatures in group II introns.

Response 1-10

We have changed the angle names as follows: former angles α , β , γ , and δ are now indicated as *angles A, B, C, and D*, respectively.

7. I strongly suggest including a cartoon panel in Fig. 1 to give the readers an overall picture of the D1 domain and where should they pay attention to. It is very hard to follow all the angle details in the current ribbon diagram.

Response 1-11

We have added a new panel (panel B) in **Figure 1**, displaying the relevant angles and distances in cartoon form.

8. I think all figure panel should be described in the main text. For example, the authors described Fig. 1G-H directly after Fig. 1A, are panels B-F not important? If so why are they in the main figure anyway? This problem exists in almost all figures.

Response 1-12

We have now re-ordered the Figures in order of appearance and ensured that all Figures are references in the main text.

9. In my opinion, the authors presented this study in a rather tedious way. For example in Fig.1, the only important message seemed to be just comparison of D1 domains in the closed and catalytic states showed $i1-i2$ helix clashed with D5. What's the point of measuring all those angles?

Response 1-13

We thank the reviewer for this comment and refer to our Response 1-3 above for rationalizing the choice of the specific angles and distances used in our manuscript.

10. If Supplementary figure 16 appears so early, maybe it should be Supplementary figure 5?

Response 1-14

We have now re-ordered the Figures in order of appearance and ensured that all Figures are references in the main text.

11. While it is glad to see that authors are aware of cryo-EM density as Coulomb

potential rather than electron density, this does not need to be repeated throughout the manuscript. “Density” is simple and accurate enough.

Response 1-15

We have changed the text according to the reviewer’s suggestion, for instance on page 10, lines 11-22; on page 25, line 21 and in the Figure legends of **Figure 3**, **Figures 5-6**, **Supplementary Figures 8**, and **Supplementary Figures 11-13**.

12. Page 2, line 6: Change molecular simulations to molecular dynamics simulations;

Response 1-16

We have changed the text “molecular simulations” to “molecular dynamics simulations”, now on page 2, lines 8-9.

Page 34, Fig 6D: Change D1-3 “fully open” to D1-3 “partly open”; Page 42-44: There are some terms not unified in the 3 Cryo-EM schematics. Also, the figures failed to include how the maps were generated after the 3DVA step;

Response 1-17

We have now relabeled **Figure 6** and corrected **Supplementary Figures 5-7** according to the reviewer’s suggestions.

Page 46, SFig 9B: duplicated residue 121.

Response 1-18

We have removed the duplicated residue, now in **Supplementary Figure 14**.

There are multiple “D12”, which is probably “D1-2”?

Response 1-19

We have corrected the spelling of D1-2 in all Figures.

Responses to reviewer #3:

Jadhav et al describe in their manuscript a detailed study on the domain assembly of a group II intron ribozyme, a most complex and large RNA machinery. Self-splicing group II introns are closely related and the evolutionary ancestors of the eukaryotic splicing machinery. From this perspective the topic is highly important not only for ribozyme specialists but for a much wider community and hence merits publication in Nature Communication.

We would like to thank the reviewer for the appreciation of our work.

Nevertheless, we find several major issues with this manuscript, which should be addressed in detail, before this manuscript could be considered for publication in Nature Commun:

Generally, the manuscript is tailored very strongly towards experts in group II intron biochemistry. Those readers not so familiar with group II introns will have problems following the argumentation and judging the results in the correct way. For example: the study does not provide information on the catalytic activity of the constructs used (D1-2, D1-3, and D1-4), while only the D1-5 construct is known to be catalytically active. Including this information, either through experimental validation or literature evidence, would strengthen the interpretation of the observed folding pathway.

Response 3-1

We would like to thank the reviewer for raising this point. Constructs D1-2, D1-3 and D1-4 are catalytically inactive because they lack the active site residues and the substrate splice junctions. To address this comment of the reviewer, we have now added this information explicitly on page 8, lines 5-7.

This is also reflected by the authors going very much into detail in the text with distances between specific nucleotides/regions, or angles between helices, etc, which makes it rather difficult and tedious to follow the overall story. The amount of detail is appreciated, but the authors might find a better way, to provide the reader with the numbers without including them in every sentence. Maybe they could also concentrate on the most important findings, adding further details in the Supplementary Material.

Response 3-2

We thank the reviewer for these considerations. In response to this comment and to comments of reviewers 1, 4, and 5, we have extensively reorganized and simplified our manuscript, for instance in the Introduction (page 3, lines 8-28; and page 4, lines 8-12); in Results (page 5, lines 33-37; page 6, lines 1-7 and lines 9-33; page 7, lines 15-32; page 8, lines 23-25; page 9, lines 8-15; page 10, lines 23-34; and page 11, lines 10-23); and in Discussion (page 14, lines 2-9, lines 15-18, and lines 21-37; page 15, lines 1-3, lines 6-14, and lines 17-34; and page 16, lines 10-21).

By SEC-SAXS that authors have experimentally and convincingly verified the overall size of the structures obtained by MD simulations. However, they also go into much detail discussing conformations of single nucleotides, solely based on simulations, and without experimental evidence (e.g. mutational analysis or others), for example on page 6/middle paragraph, but also elsewhere.

Response 3-3

To address this reviewer's comment and consolidate our mechanistic model, we have now designed new mutants of hinge 1 (residues 71-73 and 114-116) of the *O. ihayensis* group II intron that are at the core of our mechanistic model, because they regulate the conformational dynamics of the D1c helix, and thus the opening and closing of the D1 cavity. Our mutational

design aimed to rigidify the hinge, reducing its dynamics. Our hypothesis is that if the D1c dynamics are important for the assembly of group II introns into functional 3D structures, then these mutations should have an effect on catalysis, too. We note that previous mutagenesis of residues in and around hinge 1 – which had been designed in the absence of high-resolution 3D structures and which aimed at exploring the importance of nucleotide identity in hinge 1 – could not capture functional defects (Waldsich & Pyle, 2007). Here, we have successfully cloned, transcribed and purified these mutants, which – differently from previous studies – probe the importance of the secondary structure of hinge 1, rather than the corresponding nucleotide sequence. We have tested the effects of our mutations on catalysis using established enzymatic assays for this model system. In line with our mechanistic hypothesis, we could now show that rigidification of hinge 1 has a major effect on catalysis, causing significant defects in the first step of splicing (the mutant is 9-times less active than wild type) and completely impairing the second step of splicing. Importantly, we could also observe the similar effects by mutating homologous nucleotides (residues 77 and 124-126) in the *ai5y* group IIB intron from *Saccharomyces cerevisiae*. These results strongly corroborate our structural observation and support the notion that the mechanism of group II intron assembly is conserved in evolution. We have included these data in a new paragraph at the end of the Results section (page 12, lines 25-37 and page 13, lines 1-15). Consistently, we have introduced a new Figure (**Figure 8**) and we have updated the Material and Methods section with the relevant details to describe cloning and purification of the mutants, and enzymatic assay procedures (page 19, lines 3-21 and lines 24-25; page 20, lines 35-37; and page 21, lines 1-33). Finally, we have included two new authors who have performed the mutagenesis, purification, and functional assays of the wild type and mutated *ai5y* intron (Spandan Saha and Dr. Auriane Rakitch).

p 12. The energies obtained by FES for the "partly open" and "fully open" states (2 and 4 kcal·mol⁻¹). How meaningful are these energy values? What do they mean in terms of an equilibrium/distribution between these two states? The authors only talk about these two minima in the energy landscape but not on the transition in-between them, i.e. the activation barrier. Hence, the statement "This rather flat FES indicates that conformational transitions are energetically inexpensive ..." is actually such not correct, because the transition is not dependent on the values of the minima, but on the height of the activation barrier in between. In fact, as the authors actually find both states in the cryoEM, this indicates, that the barrier is actually rather high. Please modify accordingly.

Response 3-4

We agree with the reviewer on the importance of the activation barrier between the "open" states. We have now specified on page 12, lines 7-17 that the energy barrier between the "partly open" and "intermediate" states is of 6-8 kcal·mol⁻¹, and between the "intermediate" and "fully open" states of 8-10 kcal·mol⁻¹. These barriers are considerably smaller than typical energy barriers observed for conformational transitions required to refold kinetically trapped RNA conformers (Altman & Guerrier-Takada, 1986; Fedor & Uhlenbeck, 1990; Kao & Crothers, 1980). Moreover, the conformational transitions between "partly open" and "intermediate" vs "intermediate" and "fully open" states are energetically equally demanding, which explains the fact that all the minima are observed during our enhanced sampling simulations. These findings are consistent with cryo-EM data, from which the "partly open" and "fully open" states are resolved as the principal states, but are connected by a continuum of intermediate conformations, as identified by the 3DVA analysis. In other words, the energy landscape features distinct basins that are connected by accessible pathways, enabling D1 to sample a wide range of dynamic "open" conformations when D2, D3, and D4 are present. Finally, we would like to clarify that our reported free energy values – i.e. the difference of ~2 kcal·mol⁻¹ between the "partly open" and "fully open" states, and of ~4 kcal·mol⁻¹ between the "partly open" and the intermediate state – are semiquantitative estimates obtained from our metadynamics simulations. The equilibrium constant (K) is related to the free energy

difference (ΔG) between the two states by:

$$\Delta G = -RT \ln K$$

where R is the gas constant and T is the temperature.

Discussion: The discussion is very weak, as it fails to connect the findings to the larger RNA world. It lacks depth and needs to be placed in a broader context—what do the results imply for the overall folding pathway? Are the newly discovered conformations essential intermediates? How might these findings relate to other introns or RNAs, like the spliceosome? Additionally, highlighting open questions and providing a more global perspective will enhance the discussion and demonstrate its relevance to a wider scientific community.

Response 3-5

To address this reviewer's comment we have extensively modified the Discussion section of our manuscript.

We discuss the implications of our results for RNA folding on page 14, lines 2-9 and on page 17, lines 11-26. We then discuss that the newly discovered "open" intermediates are absolutely essential for group II intron folding (page 15, lines 17-28). Finally, we outline fundamental paradigms revealed by our work and open questions in the field on page 14, lines 21-37; on page 15, lines 1-14; on page 17, lines 1-4; and on page 17, lines 11-26.

Finally, we would like to note that – differently from catalysis, which is conserved between group II introns and the spliceosome – it is not possible to directly translate our findings on group II intron assembly to inform about spliceosome assembly. Indeed, in the spliceosome, assembly is mediated by a more complex process involving many protein subunits too. Although, group II intron active site is conserved in eukaryotic spliceosome, but the assembly processes of the active site or the spliceosome is entirely different. As opposed to group II introns, the spliceosome is a large RNP complex, and its assembly on the substrate pre-mRNA include multiple assembly steps that are tightly controlled by multiple proteins e.g. helicase Brr2 (Matera & Wang, 2014).

Further points in the Discussion:

First sentence (p13, l2-4): The statement that the data visualizes for the first time the multi-domain assembly pathway while "avoiding unproductive kinetic traps" should be further discussed and/or revised. While cryo-EM is indeed a powerful technique for studying RNA folding and revealing kinetic traps (Bonilla, Science Advances, 2022), the absence of observed misfolded structures does not necessarily indicate the absence of kinetic traps. Several factors may contribute to the lack of detected misfolded states, including the transient nature of kinetic traps, conformational heterogeneity and dynamics, effects of vitrification during sample preparation, resolution limitations, and particle selection bias. Given these considerations, these limitations need to be discussed in more detail to reflect the possibility that undetected misfolded states may still be present.

Response 3-6

We thank the reviewer for their comment. Our statement is motivated by a number of previous biochemical studies, which had established that group II intron does not follow a kinetically trapped mechanism (Fedorova & Zingler, 2007; Pyle et al., 2007; Su et al., 2003; Su et al., 2005; Swisher et al., 2001; Swisher et al., 2002; Waldsich & Pyle, 2007). Building on such studies, our work provides a validation to those mechanistic and biochemical observations, and shows at the molecular level the details behind the group II intron assembly process (see page 15, lines 28-37 and page 16, lines 1-9).

Two main paradigms emerge from our work and are likely of general relevance for RNAs that avoid kinetic traps.

- 1) The assembly of peripheral domains onto a pre-assembled scaffold needs to be

precisely organized and hierarchically regulated. We now develop on this point on page 14, lines 21-37 and page 15, lines 1-5.

- 2) The pre-assembled scaffold needs to be dynamic and able to sample a wide range of conformations separated by low energy barriers. We now develop on this point on page 15, lines 6-28.

Furthermore, we have now added specific considerations on page 16, lines 10-21 to warn the readers that grid vitrification may affect the overall behavior of the sample in the free-standing vitreous ice, because it may induce particle unfolding, aggregation and non-specific ice features e.g. leopard ice, thick ice or crystalline ice. Moreover, in the same paragraph, we have also stated that specific buffer conditions used in cryo-EM sample preparation may enrich or destabilize RNA conformations. Here, we importantly point out to the reader that we have been preparing our samples under conditions that ensure enzymatic activity. This strategy ensures that we sample as close to native and functionally-relevant conformations as possible (Fedorova & Zingler, 2007), as also corroborated by the fact that our structure-based mutation on hinge 1 cause catalytic defects (page 12, lines 25-37 and page 13, lines 1-15).

p14, l3-4). This sentence does not fit well in its current position, as the following sentence does not discuss other RNAs or how the insights gained can be applied more broadly. It would be more appropriate in the next section (same page, line 28), where it aligns better with the context.

Response 3-7

We agree with the reviewer and have moved the corresponding sentence as suggested, now on page 17, lines 11-12.

p14, l5 – 27: It is unclear what the two newly visualized states, "partly open" and "fully open," represent in the context of RNA folding. A more detailed discussion and interpretation is needed. Are both states essential intermediates in the folding pathway? Under what conditions would one state be favored over the other? Do the two intermediates serve distinct functional roles?

Response 3-8

To address this comment of the reviewer, we have changed the text in the Discussion on page 14, lines 21-37 and on page 15, lines 1-28. Our newly observed states are essential intermediates in the intron folding pathway because they widen the D1 pocket to receive the active site elements, following a "nutcracker" model (page 15, line 19). Intuitively, one would open a nutcracker widely to comfortably insert the nuts before cracking them. Similarly, the group II intron scaffold opens widely to recruit the active site and splice junctions in its core, before splicing (page 15, lines 17-34). The identification of these intermediates help explain a previously identified conundrum in group II intron folding, i.e. the combination of their slow folding rate (1 min^{-1}) (Swisher et al., 2002) and the lack of kinetic traps (Swisher et al., 2001; Swisher et al., 2002). We speculate that establishing a conformational equilibrium between a "partly open" and a "fully open" state is a useful strategy that group II intron adopt to control the slow formation of long-range tertiary contacts between regions located far apart from each other in the secondary structure (Swisher et al., 2002). The small but not negligible energy barriers that we capture by MD simulations (see Response 3-4) may be functional to coordinate these steps. This observation is also in line with a number of biochemical observations from chemical probing experiments (page 15, lines 34-37 and page 16, lines 1-9).

p14, l9 - 22: The description of the D1 transition is unclear.

- a) The text states that multi-domain assembly is not accomplished through sequential conformational changes of D1, INSTEAD, D1 transitions from a closed state to partly open and fully open. The latter suggests a sequential pathway, so there is

no contradiction - please elaborate on this point?

Response 3-9

We thank the reviewer for this comment. We would like to clarify that domain assembly process is indeed sequential: D2 docks on D1, followed by D3, then by D4, then by D5. But the conformations that D1 explores throughout this process are not sequential. From a “closed” state, D1 does not open up gradually towards the “catalytic” state. Instead, it first opens up more widely than the “catalytic” state and forms an equilibrium of open states (ranging from the “partly” to the “fully” open states). Only then, D1 closes up again into the “catalytic” state. We have now rephrased this concept on page 14, lines 24-37 and page 15, lines 1-14.

- b) However, figure 7 indicates that D1 can adopt either the partially open or fully open state, raising the question of which description is now correct. Does the transition take place sequentially or one after the other?

Response 3-10

We thank the reviewer for this comment. The “partly open” and “fully open” states are in equilibrium representing the two most extreme conformations explored by D1, while sampling a continuum of conformations as captured by our 3D variability analysis (**Figures 4-5, Supplementary Figures 5-7, Supplementary Figure 9**). In **Figure 9**, we now reflect such equilibrium by displaying a middle vertical double-pointed arrow in wheat color in the center of the figure.

- c) Furthermore, if D1 can adopt either the partially or fully open state, what determines which conformation is formed? And if the two open states are interchangeable, what is the trigger for this transition?

Response 3-11

In response to this comment of the reviewer, we would like to confirm that the “open” states are interchangeable and in a continuum equilibrium. Our MD simulations show that conformational transitions along this conformational landscape have low energy barriers (see Response 3-4). We have explicitly added these considerations in the main text on page 12, lines 7-17 and on page 15, lines 10-14.

p14, 25 – 27: Also here, please elaborate more on this dynamic gating system with regard to a large D4. Describe how D1 should look like so that D4 can be proper docked and the ORF is not interfering

Response 3-12

We have addressed this comment of the reviewer by modifying the text on page 14, lines 21-37 and page 15, lines 1-5. Briefly, our mechanistic model, which shows that D4 points out of the D1 core during group II intron assembly, is valid independent of the length of D4. D4 does not need to dock on D1 at any stage of the assembly process, and indeed our structures show that it is efficiently excluded from the core of the ribozyme at all stages, leaving room for D5 to enter the active site as soon as available.

The Video is well made and effectively helps to visualize the process. However, in some places, additional labelling would improve clarity, and some transitions happen too quickly, making it difficult to follow. Below are specific suggestions for improvement:

Response 3-13

We have updated **Movie 1** according to the suggestions of the reviewer.

Beginning: Indicate which conformation is being shown.

Response 3-14

We now mention which conformation we show in the beginning of the video.

Second 6: Clarify what the yellow conformation represents.

Response 3-15

We now mention that the yellow conformation represents D1 in its “catalytic” state.

Second 16: This transition is too fast and should be slowed down.

Response 3-16

We have now slowed the transition at second 16 of the video.

Second 34: Label the blue domain for clarity.

Response 3-17

The blue domain is now labelled accordingly.

Second 38: Specify which conformation the blue Coulomb potential density corresponds to, and indicate which constructs are being shown.

Response 3-18

This suggestion is now implemented in the video.

Second 70: Label the purple domain and the later-appearing purple density.

Response 3-19

The purple domain is now labelled in the video.

Second 70: Clearly indicate which constructs the purple structures correspond to (D1-3)

Response 3-20

We now mention the names of the construct in the updated video.

Second 104: Label the orange construct (D1-4) for consistency.

Response 3-21

We have now labelled the orange construct as D1-4.

Adding these details would enhance the video's clarity and ensure that viewers can fully understand the structural changes being depicted.

Response 3-22

We thank the reviewer for their useful suggestions.

Comments to Figures:

Figure 1a: (i) Since this manuscript is intended for a broader audience beyond the group II intron community, the structural deviations from the wild-type construct must be mentioned/shown, i.e. the absence of D6 and the truncation of D2, D3, and D4. In the end, the authors are using a heavily modified construct and this will help readers unfamiliar with these modifications better understand the construct.

(ii) Please indicate the two exon binding sites and clarify whether the intron binding

site is located on the 5' exon to provide additional structural context.

Response 3-23

We have modified **Figure 1** according to the reviewer's suggestions.

Supplementary Figure 4 – SEC Profile: (i) C-F: Including the calculated molecular weight (MW) of each construct would help clarify the interpretation of the mass photometry data.

ii) C-F: The presence of two peaks in the SEC profiles is not explained—please provide an explanation for this observation.

(iii) In panels 4C and 4D, the main peak appears to have the same MW despite representing two different constructs. Could this be an error, or is there a specific reason for this result? Clarification would be appreciated.

Response 3-24

We thank the reviewer for these suggestions, but would like to politely point out that panels C-F in this figure represent mass photometry profiles, not size-exclusion chromatography profiles. This said, the reviewer's suggestion is well taken, and we have modified the figure (now **Supplementary Figure 2**) accordingly.

First, we now indicate the theoretical molecular weight of each construct next to the mass photometry data (panels C-F).

Second, we discuss the profile of the mass distribution plot in the legend of **Supplementary Figure 2**. Briefly, we explain that in the mass photometer experiment, we specifically track the binding of the molecules to the glass surface. During these binding events, two molecules may bind the surface very close to each other. This could give rise to a secondary large-molecular-weight peak observed in the mass distribution plot. Based on these considerations and on the results from native gel electrophoresis (**Supplementary Figure 2**, panel A), size-exclusion chromatography (**Supplementary Figure 2**, panel B), SAXS (**Supplementary Figure 3**), and cryo-EM (**Supplementary Figures 5-7**) – all showing monomeric forms of our constructs – we suggest that the secondary larger-molecular-weight peak is a technical artifact.

Third, we have updated **Supplementary Figure 2**, panel D, with the correct mass distribution plot for construct D1-2, and apologize for the mistake in the previous draft.

Supplementary Figure 9: (i) There is an inconsistency between the text (p.15, l.17), which refers to "hinge 1," and the figure, which labels "hinge 2." Please clarify which is correct and ensure consistency.

Response 3-25

We thank the reviewer for noticing this typo. We have changed the figure (now **Supplementary Figure 14**) accordingly.

Responses to reviewer #4:

The authors combine single-particle cryo-em, SAXS, and molecular dynamics simulations to illustrate the multi-domain assembly of a self-splicing group II intron – a system of considerable biomedical significance. The work is highly innovative and addresses a fundamental problem of multi-domain assembly of RNA into a functional state. Specifically, the work sheds light on important intermediates in the assembly pathway to achieve the catalytic state, and generally demonstrates how RNA domains may strategically assemble to avoid kinetic traps. The provided movie is especially helpful in understanding this complex process. This work constitutes a large-scale interdisciplinary collaboration that is, to my knowledge, completely novel in the insights that have been gained. The work is likely to be of broad interest to the community and readership of Nature journals.

We would like to thank the reviewer for the appreciation of our work.

There are a few clarifications and concerns that should be addressed before publication can be recommended:

1. A clearer description of how different structural data was used throughout this study is needed. In the paper, the authors compare geometric properties of crystal structures of the "closed" state (PDB ID 4Y1O) and "catalytic" state (PDB ID 4FAQ) from the referenced 2012 Cell paper. 4FAQ was crystallized with Ca²⁺ and it forms a dimer in the crystal. 4Y1O was crystallized with Mg²⁺ and is a monomer in the crystal. The following is my understanding of what the authors did (but it was not completely clear due to lack of details). Simulations depart from 4FAQ (catalytic) and do steered MD to get the "closed" state structure. The simulations are done with Mg²⁺, but supposedly depart from a structure containing Ca²⁺ with no description of these being replaced by Mg²⁺ or how that was achieved. Later in the paper the authors mention the catalytic state as 4FAR instead of 4FAQ, but it could be a typo because that is the only instance I saw it mentioned. That said, 4FAR is a structure from the same 2012 paper, but it is crystallized with Mg²⁺ and is a monomer. This does seem like a more appropriate structure to make comparisons with, so it left me questioning whether which structure they used in which instance. I think this illustrates that the manuscript would benefit with a clear summary of what structure(s) were used in what instances, and to provide additional details of the simulation set up. Some of the comments below are based on my assumption of what the authors did.

Response 4-1

We thank the reviewer for this comment and for their insightful considerations regarding the structural data used in our study. We have now expanded our methods section to provide a clearer summary description of the structures used in MD, including a detailed description of our simulation setup and ion replacement protocol, on page 21, lines 35-36 and page 22, lines 1-19. We confirm that the "catalytic state" structure we used for our MD simulations in our initial draft corresponds to PDB ID 4FAQ, not 4FAR (we apologize for the typo in our original draft). Our choice of 4FAQ over 4FAR was deliberately motivated by the fact that 4FAQ was crystallized with Ca²⁺ ions, which support the intron's structural folding but not catalysis and thus captures a pre-catalytic state of the ribozyme. In 4FAQ, the 5' exon remains ligated to the intron, mirroring the "closed" state (4Y1O). In contrast, 4FAR, crystallized with Mg²⁺, represents a post-catalytic structure where the 5' splice site has undergone hydrolytic cleavage. Importantly, structures 4FAQ and 4FAR superpose closely, with an overall all-atom RMSD of only 0.6 Å (Marcia & Pyle, 2012). Moreover, structural and functional ions are conserved across 4FAQ, 4FAR, and also 4Y1O (Marcia & Pyle, 2012; Zhao et al., 2015). This conservation likely explains why it is possible to replace Ca²⁺ ions with Mg²⁺ ions in these structures without affecting the behavior of the molecules in our MD simulations and thus the corresponding results, as we have already extensively verified and confirmed in one of our

previous, recently-published study (Silvestri et al., 2024). We have now added these important considerations and related supporting literature to our manuscript on page 22, lines 9-19. It is based on these considerations that, for our simulations, we initiated steered-MD from the 4FAQ structure with Mg^{2+} ions to model the transition to the "closed" state. In this revision, though, we also present a new simulation that constructively addresses the reviewer's concern and that helps to further strengthen our mechanistic results. This new simulation is an equilibrium MD simulation, where we have used our modelled D1 in isolation (i.e., D1 coordinates from 4FAQ, Ca^{2+} replaced with Mg^{2+}). Our simulations show that in these conditions, D1 spontaneously collapses into closed conformations similar to those found in 4Y1O. This result suggests that replacing Ca^{2+} with Mg^{2+} reproduces the behavior of D1 in isolation (**Supplementary Figure 4**, new panel A).

2. Page 5: The authors compare angles and displacements between structures of the "closed" and "catalytic" states from PDB's 4Y1O and 4FAQ. The 4Y1O (closed) structure was crystallized as a dimer, while the 4FAQ (catalytic) structure was crystallized as a monomer. Are the authors concerned that crystal packing interactions present in the dimer may impact these observations?

Response 4-2

We do not think crystal packing in structure 4Y1O would impact our observations because the hinge 1 residues that control the conformational flexibility of D1 – according to our cryo-EM and MD simulations – are not involved neither in dimer formation nor in crystal contact formation. The residues that mediate crystal contacts in 4Y1O are 180-181, 223, 267-270, far away from the motifs that we characterize in our present study.

3. Page 9 line 29: The catalytic conformation is listed as PDB ID 4FAR. All other instances are 4FAQ. Is this correct? Both are structures of *Oceanobacillus iheyensis* group II intron but with different conditions. Does 4FAR make more sense?

Response 4-3

We thank the reviewer for noticing this typo. We have now changed 4FAR with the correct 4FAQ code, on page 9, line 24.

Regarding the choice between 4FAQ and 4FAR, we kindly direct the reviewer to Response 4-1, above. There, we provide a detailed explanation of our rationale for selecting 4FAQ as the representative structure for the catalytic conformation in our study. This choice was made deliberately, taking into account the specific characteristics and catalytic stages of both structures.

Computational details:

4. Page 18, Structure models for MD simulations: When preparing the systems (D1 etc) for SAXS-driven metadynamics metainference simulations, the authors started from the catalytic state and obtained the closed state from steered molecular dynamics. It is unclear how the placement of crystallographic metal ions, which are crucial for RNA folding and function [JPCB 126 (32), 5982-5990], were handled. The catalytic structure contains Ca^{2+} and K^+ ions while the closed structure contains Mg^{2+} and K^+ ions. Were the Ca^{2+} ion positions replaced with Mg^{2+} , or were Mg^{2+} ions randomly placed to achieve the desired concentration? To what degree are the positions of these ions conserved between the states? The placement of Mg^{2+} ions in the simulations are very important, as they are important to assembly, folding and function. Further, the model for Mg^{2+} ions used by the authors, if their description is correct, is quite poor with respect to binding properties with RNA (see below). One could make predictions about Mg^{2+} ion binding sites for a given initial structure using 3D-RISM such as in [JACS 141 (6), 2435-2445]. Could such a procedure be validated by the structure with Mg^{2+} and then applied to other states?

Response 4-4

We appreciate the reviewer's insightful comment regarding the handling of metal ions in our simulations. As mentioned in our Response 4-1, the functional cations within D1 are conserved and well-defined in both the 4Y1O and 4FAQ crystal structures (Marcia & Pyle, 2012; Zhao et al., 2015). Moreover, our structural ions are also preserved across all group II intron catalytic states (Marcia & Pyle, 2014). Given this conservation, we did not need to predict cation binding sites, as these ions were already present in our crystallographic starting structure. When performing steered MD to model the "closed" state, beginning from the 4FAQ structure, we simply replaced each Ca^{2+} atom type with Mg^{2+} . This approach allowed us to maintain the critical ion positions while transitioning to the catalytically relevant Mg^{2+} ions. We had already validated such modeling for MD simulations in previous studies (Manigrasso et al., 2020; Silvestri et al., 2024). However, as mentioned above in Response 4-1, we have now additionally run new MD simulations that confirm that that replacement of Ca^{2+} with Mg^{2+} does not affect the results of our simulations (**Supplementary figure 4**, new panel A).

This said, we fully recognize the importance of proper ion placement for RNA folding and function, as highlighted by the reviewer's reference to recent literature. Our methodological approach preserved the experimentally determined ion positions, providing a robust starting point for our simulations. We have now clarified this procedure in the methods section of our revised manuscript to ensure transparency in our approach (page 22, lines 9-19).

5. Page 19, MD simulation setup: The authors state that the TIP3P water model was used and Mg^{2+} ions were treated with parameters by Li et al. The divalent ion parameters of Li et al. were developed in bulk water and not balanced with their interactions with nucleic acids. This has been demonstrated to be important for the more advanced 12-6-4 models used with TIP4Pew water and these are what is recommended for RNA simulations, see [JPCB 119 (50), 15460-15470] (the authors might want to consider this in the future – it is important if one is making predictions about divalent metal ion binding sites). Additionally, it is not clear whether the total ion concentrations of Mg^{2+} and K^{+} (100 mM and 150mM) were balanced with counterions to produce a neutral system and what parameters were used (perhaps I missed this detail).

Response 4-5

We appreciate the reviewer's insightful comment regarding the choice of ion parameters in our simulations. While we acknowledge the recent developments in cation parameterization, we would like to offer the following rationale for our approach:

The Li and Merz parameters, when used in combination with the TIP3P water model and AMBER ChiOL3 corrections for RNA, have been demonstrated to be successful in modeling RNA-bound, structurally stable Mg^{2+} cations (Casalino et al., 2017; Manigrasso et al., 2020; Manigrasso et al., 2021). These parameters are known to have limitations in accurately modeling the kinetics of water-cation exchange at binding sites. However, for our specific study focusing on highly-structured, and steadily bound cations, this limitation is less critical, as we have previously proven (Muscat et al., 2024; Sponer et al., 2018). This said, we appreciate the reviewer's suggestion and will certainly consider using the most appropriate cation parameters for future studies, depending on the specific events under investigation. As soon as the 12-6-4 models are implemented in GROMACS, such implementation will be also much simpler from a practical standpoint.

Regarding system neutrality, we have now clarified in the manuscript that Mg^{2+} and K^{+} ions were added in a ratio of 1:3 to mimic the 50mM:150mM concentrations, as used in SAXS buffers (page 23, lines 11-14).

6. Page 12 paragraph 2: Details of system preparation for the "partly open" and "fully open" metadynamics simulations should be provided in the supporting information.

Response 4-6

We understand the concerns of the reviewer, and would like to politely point them to the comprehensive description of our system construction and the metadynamics simulation setup, which we provide in the Materials and Methods section – “Structural models for MD simulations” (pages 21-23).

We additionally acknowledge and have now corrected some confusing terminology present in our initial draft. In the main text on page 11, line 24, we referred to 'metadynamics', while in the Materials and Methods section, we specifically used the term 'well-tempered metadynamics simulations'. To address this inconsistency and improve clarity, we have now updated the terminology in the main text and decided to consistently use the phrase 'well-tempered metadynamics' (page 11, line 24). We have also added a statement directing readers to the Materials and Methods section for further details (page 11, lines 27-28).

7. SI figure 1A: The y axis label is missing a unit. Should the x axis label be residue?

Response 4-7

We have made the required change to **Supplementary Figure 1**.

Responses to reviewer #5:

We thank the reviewer for their efforts and refer them to our detailed point-by-point answer to the other colleagues.

References:

- Altman, S., & Guerrier-Takada, C. (1986). M1 RNA, the RNA subunit of Escherichia coli ribonuclease P, can undergo a pH-sensitive conformational change. *Biochemistry*, 25(6), 1205-1208. <https://doi.org/10.1021/bi00354a002>
- Bailor, M. H., Mustoe, A. M., Brooks, C. L., 3rd, & Al-Hashimi, H. M. (2011). 3D maps of RNA interhelical junctions. *Nat Protoc*, 6(10), 1536-1545. <https://doi.org/10.1038/nprot.2011.385>
- Casalino, L., Palermo, G., Abdurakhmonova, N., Rothlisberger, U., & Magistrato, A. (2017). Development of Site-Specific Mg(2+)-RNA Force Field Parameters: A Dream or Reality? Guidelines from Combined Molecular Dynamics and Quantum Mechanics Simulations. *J Chem Theory Comput*, 13(1), 340-352. <https://doi.org/10.1021/acs.jctc.6b00905>
- Cech, T. R. (2002). Ribozymes, the first 20 years. *Biochem Soc Trans*, 30(Pt 6), 1162-1166. <https://doi.org/10.1042/>
- Chen, S., McMullan, G., Faruqi, A. R., Murshudov, G. N., Short, J. M., Scheres, S. H., & Henderson, R. (2013). High-resolution noise substitution to measure overfitting and validate resolution in 3D structure determination by single particle electron cryomicroscopy. *Ultramicroscopy*, 135, 24-35. <https://doi.org/10.1016/j.ultramic.2013.06.004>
- Enkhbayar, P., Damdinsuren, S., Osaki, M., & Matsushima, N. (2008). HELFIT: Helix fitting by a total least squares method. *Comput Biol Chem*, 32(4), 307-310. <https://doi.org/10.1016/j.compbiolchem.2008.03.012>
- Fedor, M. J., & Uhlenbeck, O. C. (1990). Substrate sequence effects on "hammerhead" RNA catalytic efficiency. *Proc Natl Acad Sci U S A*, 87(5), 1668-1672. <https://doi.org/10.1073/pnas.87.5.1668>
- Fedorova, O., & Zingler, N. (2007). Group II introns: structure, folding and splicing mechanism. *Biol Chem*, 388(7), 665-678. <https://doi.org/10.1515/BC.2007.090>
- He, J., Zeil, S., Hallak, H., McKaig, K., Kovacs, J., & Wriggers, W. (2015). *Comparison of an atomic model and its cryo-EM image at the central axis of a helix* 2015 IEEE International Conference on Bioinformatics and Biomedicine (BIBM), <https://ieeexplore.ieee.org/stampPDF/getPDF.jsp?tp=&arnumber=7359860&ref=>
- Herschlag, D., Bonilla, S., & Bisaria, N. (2018). The Story of RNA Folding, as Told in Epochs. *Cold Spring Harb Perspect Biol*, 10(10). <https://doi.org/10.1101/cshperspect.a032433>
- Kao, T. H., & Crothers, D. M. (1980). A proton-coupled conformational switch of Escherichia coli 5S ribosomal RNA. *Proc Natl Acad Sci U S A*, 77(6), 3360-3364. <https://doi.org/10.1073/pnas.77.6.3360>
- Kretsch, R. C., Albrecht, R., Andersen, E. S., Chen, H.-A., Chiu, W., Das, R., Gezelle, J. G., Hartmann, M. D., Höbartner, C., Hu, Y., Jadhav, S., Johnson, P. E., Jones, C. P., Koirala, D., Kristoffersen, E. L., Largy, E., Lewicka, A., Mackereth, C. D., Marcia, M.,...Kryshtafovych, A. (2025). Functional relevance of CASP16 nucleic acid predictions as evaluated by structure providers. *bioRxiv*. <https://doi.org/10.1101/2025.04.15.649049>
- Li, S., Palo, M. Z., Pintilie, G., Zhang, X., Su, Z., Kappel, K., Chiu, W., Zhang, K., & Das, R. (2022). Topological crossing in the misfolded Tetrahymena ribozyme resolved by cryo-EM. *Proc Natl Acad Sci U S A*, 119(37), e2209146119. <https://doi.org/10.1073/pnas.2209146119>
- Lietzke, S. E., Barnes, C. L., Berglund, J. A., & Kundrot, C. E. (1996). The structure of an RNA dodecamer shows how tandem U-U base pairs increase the range of stable RNA structures and the diversity of recognition sites. *Structure*, 4(8), 917-930. [https://doi.org/10.1016/s0969-2126\(96\)00099-8](https://doi.org/10.1016/s0969-2126(96)00099-8)
- Lu, X. J., & Olson, W. K. (2003). 3DNA: a software package for the analysis, rebuilding and visualization of three-dimensional nucleic acid structures. *Nucleic Acids Res*, 31(17), 5108-5121. <http://www.ncbi.nlm.nih.gov/pubmed/12930962>

- Manigrasso, J., Chillon, I., Genna, V., Vidossich, P., Somarowthu, S., Pyle, A. M., De Vivo, M., & Marcia, M. (2020). Visualizing group II intron dynamics between the first and second steps of splicing. *Nat Commun*, *11*(1), 2837. <https://doi.org/10.1038/s41467-020-16741-4>
- Manigrasso, J., De Vivo, M., & Palermo, G. (2021). Controlled Trafficking of Multiple and Diverse Cations Prompts Nucleic Acid Hydrolysis. *ACS Catal*, *11*(14), 8786-8797. <https://doi.org/10.1021/acscatal.1c01825>
- Marcia, M., & Pyle, A. M. (2012). Visualizing group II intron catalysis through the stages of splicing. *Cell*, *151*(3), 497-507. <https://doi.org/10.1016/j.cell.2012.09.033>
- Marcia, M., & Pyle, A. M. (2014). Principles of ion recognition in RNA: insights from the group II intron structures. *Rna*, *20*, 516-527. <https://doi.org/10.1261/rna.043414.113>
- Matera, A. G., & Wang, Z. (2014). A day in the life of the spliceosome. *Nat Rev Mol Cell Biol*, *15*(2), 108-121. <https://doi.org/10.1038/nrm3742>
- Mohan, S., Donohue, J. P., & Noller, H. F. (2014). Molecular mechanics of 30S subunit head rotation. *Proc Natl Acad Sci U S A*, *111*(37), 13325-13330. <https://doi.org/10.1073/pnas.1413731111>
- Mohan, S., & Noller, H. F. (2017). Recurring RNA structural motifs underlie the mechanics of L1 stalk movement. *Nat Commun*, *8*(1), 14285. <https://doi.org/10.1038/ncomms14285>
- Morris, K. V., & Mattick, J. S. (2014). The rise of regulatory RNA. *Nat Rev Genet*, *15*(6), 423-437. <https://doi.org/10.1038/nrg3722>
- Muscat, S., Martino, G., Manigrasso, J., Marcia, M., & De Vivo, M. (2024). On the Power and Challenges of Atomistic Molecular Dynamics to Investigate RNA Molecules. *J Chem Theory Comput*. <https://doi.org/10.1021/acs.jctc.4c00773>
- Pettersen, E. F., Goddard, T. D., Huang, C. C., Meng, E. C., Couch, G. S., Croll, T. I., Morris, J. H., & Ferrin, T. E. (2021). UCSF ChimeraX: Structure visualization for researchers, educators, and developers. *Protein Sci*, *30*(1), 70-82. <https://doi.org/10.1002/pro.3943>
- Pyle, A. M., Fedorova, O., & Waldsich, C. (2007). Folding of group II introns: a model system for large, multidomain RNAs? *Trends Biochem Sci*, *32*(3), 138-145. <https://doi.org/10.1016/j.tibs.2007.01.005>
- Silvestri, I., Manigrasso, J., Andreani, A., Brindani, N., Mas, C., Reiser, J. B., Vidossich, P., Martino, G., McCarthy, A. A., De Vivo, M., & Marcia, M. (2024). Targeting the conserved active site of splicing machines with specific and selective small molecule modulators. *Nat Commun*, *15*(1), 4980. <https://doi.org/10.1038/s41467-024-48697-0>
- Sponer, J., Bussi, G., Krepl, M., Banas, P., Bottaro, S., Cunha, R. A., Gil-Ley, A., Pinamonti, G., Poblete, S., Jurecka, P., Walter, N. G., & Otyepka, M. (2018). RNA Structural Dynamics As Captured by Molecular Simulations: A Comprehensive Overview. *Chem Rev*, *118*(8), 4177-4338. <https://doi.org/10.1021/acs.chemrev.7b00427>
- Su, L. J., Brenowitz, M., & Pyle, A. M. (2003). An alternative route for the folding of large RNAs: apparent two-state folding by a group II intron ribozyme. *J Mol Biol*, *334*(4), 639-652. <http://www.ncbi.nlm.nih.gov/pubmed/14636593>
- Su, L. J., Waldsich, C., & Pyle, A. M. (2005). An obligate intermediate along the slow folding pathway of a group II intron ribozyme. *Nucleic Acids Res*, *33*(21), 6674-6687. <https://doi.org/10.1093/nar/gki973>
- Swisher, J., Duarte, C. M., Su, L. J., & Pyle, A. M. (2001). Visualizing the solvent-inaccessible core of a group II intron ribozyme [Research Support, U.S. Gov't, P.H.S.]. *Embo J*, *20*(8), 2051-2061. <https://doi.org/10.1093/emboj/20.8.2051>
- Swisher, J. F., Su, L. J., Brenowitz, M., Anderson, V. E., & Pyle, A. M. (2002). Productive folding to the native state by a group II intron ribozyme. *J Mol Biol*, *315*(3), 297-310. <https://doi.org/10.1006/jmbi.2001.5233>
- Thangappan, J., Wu, S., & Lee, S. G. (2017). Joint-based description of protein structure: its application to the geometric characterization of membrane proteins. *Sci Rep*, *7*(1), 1056. <https://doi.org/10.1038/s41598-017-01011-z>
- Topf, M., Maiorca, M., Jadhav, S., Sweeney, A., Marini, G., Mulvaney, T., & Marcia, M. (2025). Uncovering Hidden Functional States in Cryo-EM Datasets with EMPROVE. *Research Square*. <https://doi.org/10.21203/rs.3.rs-6535769/v1>

- Waldsich, C., & Pyle, A. M. (2007). A folding control element for tertiary collapse of a group II intron ribozyme. *Nat Struct Mol Biol*, 14(1), 37-44. <https://doi.org/10.1038/nsmb1181>
- Zhao, C., Rajashankar, K. R., Marcia, M., & Pyle, A. M. (2015). Crystal structure of group II intron domain 1 reveals a template for RNA assembly. *Nat Chem Biol*, 11(12), 967-972. <https://doi.org/10.1038/nchembio.1949>

RESPONSE TO REVIEWERS' COMMENTS FOR:

Dynamic assembly of a large multidomain ribozyme visualized by cryo-electron microscopy

Shekhar Jadhav^{1,7}, Mauro Maiorca^{2,3,4,#}, Jacopo Manigrasso^{5,6,#}, Spandan Saha⁷, Auriane Rakitch⁷, Stefano Muscat⁵, Thomas Mulvaney^{2,3,4}, Marco De Vivo^{5,*}, Maya Topf^{2,3,4,*}, Marco Marcia^{1,7,8,9,*}

*To whom correspondence should be addressed.

E-mail: marco.devivo@iit.it; maya.topf@cssb-hamburg.de; marco.marcia@icm.uu.se

We would like to thank the editor and all the reviewers for their careful assessment of our manuscript and for their comments and criticism. In this detailed point-by-point response, we have addressed all remaining concerns of reviewer 1. We enclose a revised version of the manuscript and figures, in which revisions are marked in red.

Responses to reviewer #1:

The revised manuscript by Jadhav, et al. is substantially improved. Including additional biochemical validation of hinge 1 mutagenesis is informative. The revised content is still somewhat hard to follow. The figure panels are not included in all figure citations in the text, making the entire manuscript hard to focus and follow.

Response 1-1

We have now included panel information in all relevant figure citations in the text.

Initially, my major problem is understanding the logical connection between structures of gradually longer constructs (D1, D1-2, D1-3, D1-4) and the sequential order of this assembly, as described in P10 Line 7 – 34. Later in the Discussion in P14 Line 28-36, this seems to be a reasonable explanation, which I strongly suggest to move to the corresponding paragraph in the Results section. The Discussion can simply keep the “first come, first fold”.

Response 1-2

We have moved the text – former on page 14, lines 28-36 – to the results section, currently on page 10, lines 27-35.

SAXS measures structural features in solution, which is indeed complementary to cryo-EM. However, I am not convinced that SAXS-driven simulations initiate or validate the working hypothesis of this study. SAXS results here represent the average globular shapes of different RNA constructs in solution. I cannot imagine the SAXS results indicate the presence of “closed”, “partly open” and “fully open” conformations that facilitate subsequent MD simulations and high-resolution cryo-EM analysis. Neither do I acknowledge that SAXS results validate the existence of these conformations in solution. Maybe the X2 values mentioned by the authors can be correlated to distribution of different conformations, but unless explicitly explained in the main text, I suggest to leave SAXS in the SI simply as in-solution results.

Response 1-3

We fully agree with the reviewer. Our SAXS data do not indicate the presence of “closed”, and of “partly/fully open” states. Instead, SAXS data importantly confirm that purified constructs are homogeneous and folded in solution, as described on page 6, lines 17-20, and on page 7, lines 18-20. What instead triggers the initial hypothesis of our study, i.e. that group II intron scaffold domain D1 may explore “open” conformations in the presence of downstream domains, are the results from SAXS-driven metainterference metadynamics simulations, as

reported on page 7, lines 35-36. Thus, we feel that our current draft is indeed already in line with the suggestions and comments of the reviewer: the MD data are reported in the main text figures (Figure 2), whereas the SAXS data are reported in the Supplementary Information File (Supplementary Figure 3 and Supplementary Tables 2-3), as suggested by the reviewer.

Energy barriers between conformations are interesting topics, the authors may consider including relevant literatures (e.g. <https://doi.org/10.1038/s41467-022-29332-2>) discussing the energy landscape preserved by cryo-EM. These energy numbers, although estimated, could explain why such conformations are resolved.

Response 1-4

We have included the suggested reference, now on page 12, line 23.

Other minor comments:

1. P3 Line 6 should probably name what are the two main pathways?

Response 1-5

We have added the names of the two RNA structural assembly pathways on page 3, line 7.

2. labelling the critical hinge 1 and 2 in Fig 1C-G helps the readers to visualize them in 3D.

Response 1-6

We have modified Figure 1C-G accordingly, and updated the corresponding Figure legend.

3. P5 Line 13 – 19: When claiming the “least” structurally-displaced residues, is it actually that both structures were aligned to D1d1? This should be specified.

Response 1-7

We now explain on page 5, lines 16-18, that we have performed an all-atom superposition of D1 coordinates of the “closed” D1 and the “catalytic” D1 using PyMOL.

4. P5 Line 24: I am guessing the clash is referring to Fig. 1I, can this clash be highlighted so that readers know what to look at.

Response 1-8

We have highlighted the steric clash between A72 and the Z-anchor and λ motifs in Figure 1I, and updated the corresponding Figure legend.

5. P6 Line 7: Make the right panel of Fig. 1G a separate panel, as only this panel contains D5, and highlight the clash between D5 and closed i1-i2, so that readers know what to look at. Cite the movie as well.

Response 1-9

In Figure 1, as recommended, we have produced a separate panel 1H highlighting the clash between D5 and closed i1-i2, and we have updated the corresponding Figure legend. We have also added a reference to the movie on page 6, line 11.

6. Fig 6A-D: These panels are trying to highlight the density supporting C116 and A72, but it's really not clear what should be looked at. Are the dashed circles suggesting missing density, which corresponds to bulged out conformation?

Response 1-10

We have now improved Figure 6 as recommended, and updated the corresponding Figure legend.

Responses to reviewer #3:

Jadhav et al have extensively revised their manuscript according to the reviewers comments. My previous comments and issues have now all been attended to and sufficiently clarified. As stated previously, the topic is highly important not only for ribozyme specialists but for a much wider community, and hence I support now publication in Nature Communication.

Response 3-1

We would like to thank the reviewer for the constructive effort in stimulating us to improve our manuscript.

Responses to reviewer #4:

The authors have adequately addressed to concerns raised in the previous review and revised the manuscript accordingly.

Response 4-1

We would like to thank the reviewer for the constructive effort in stimulating us to improve our manuscript.